# African Swine Fever Virus MGF110-7L Induces Host Cell Translation Suppression and Stress Granule Formation by Activating the PERK/PKR-eIF2$\alpha$ Pathway

Han Zhong,[a] Shuai Fan,[a] Yongkun Du,[a,b] Yuhang Zhang,[a] Angke Zhang,[a] Dawei Jiang,[a,b] Shichong Han,[a] Bo Wan,[a,b] Gaiping Zhang[a,c]

[a]International Joint Research Center of National Animal Immunology, College of Veterinary Medicine, Henan Agricultural University, Zhengzhou, Henan, People's Republic of China

[b]Henan Engineering Laboratory of Animal Biological Products, Henan Agricultural University, Zhengzhou, Henan, People's Republic of China

[c]Longhu Laboratory, Zhengzhou, Henan, People's Republic of China

Han Zhong and Shuai Fan contributed equally to this article. The order was determined by the corresponding author after negotiation.

**ABSTRACT**   African swine fever (ASF) is a highly contagious and often lethal disease of pigs caused by ASF virus (ASFV) and recognized as the biggest killer in global swine industry. Despite exhibiting incredible self-sufficiency, ASFV remains unconditionally dependent on the host translation machinery for its mRNA translation. However, less is yet known regarding how ASFV-encoded proteins regulate host translation machinery in infected cells. Here, we examined how ASFV interacts with the eukaryotic initiation factor 2$\alpha$ (eIF2$\alpha$) signaling axis, which directs host translation control and adaptation to cellular stress. We found that ASFV MGF110-7L, a previously uncharacterized member of the multigene family 110, remarkably enhanced the phosphorylation level of eIF2$\alpha$. In porcine alveolar macrophage 3D4/21 and porcine kidney-15 cells, MGF110-7L triggered eIF2$\alpha$ signaling and the integrated stress response, resulting in the suppression of host translation and the formation of stress granules (SGs). Mechanistically, MGF110-7L-induced phosphorylation of eIF2$\alpha$ was mediated via protein kinase R (PKR) and PKR-like endoplasmic reticulum (ER) kinase (PERK), and this process was essential for host translation repression and SG formation. Notably, our subsequent analyses confirmed that MGF110-7L was overwhelmingly retained in the ER and caused a specific reorganization of the secretory pathway. Further proteomic analyses and biochemical experiments revealed that MGF110-7L could trigger ER stress and activate the unfolded protein response, thus contributing to eIF2$\alpha$ phosphorylation and translation reprogramming. Overall, our study both identifies a novel mechanism by which ASFV MGF110-7L subverts the host protein synthesis machinery and provides further insights into the translation regulation that occurs during ASFV infection.

**IMPORTANCE**   African swine fever (ASF) has become a socioeconomic burden and a threat to food security and biodiversity, but no commercial vaccines or antivirals are available currently. Understanding the viral strategies to subvert the host translation machinery during ASF virus (ASFV) infection could potentially lead to new vaccines and antiviral therapies. In this study, we dissected how ASFV MGF110-7L interacts with the eIF2$\alpha$ signaling axis controlling translational reprogramming, and we addressed the role of MGF110-7L in induction of cellular stress responses, eIF2$\alpha$ phosphorylation, translation suppression, and stress granule formation. These results define several molecular interfaces by which ASFV MGF110-7L subverts host cell translation, which may guide research on antiviral strategies and dissection of ASFV pathogenesis.

**KEYWORDS**   African swine fever virus, MGF110-7L, eIF2$\alpha$ phosphorylation, translation suppression, stress granule, endoplasmic reticulum stress

Address correspondence to Shichong Han, hanshichong081@126.com, Bo Wan, wanboyi2000@163.com, or Gaiping Zhang, zhanggaip@126.com.

The authors declare no conflict of interest.

African swine fever (ASF) is a highly contagious disease in domestic and wild pigs, whose mortality rate can reach 100%. ASF was first reported in Kenya in 1921, and now it has spreaded to more than 70 countries in five continents, African, Europe, Asia, Oceania, and Americas (OIE, www.oie.int/en/animal-health-in-the-world/animal-diseases/african-swine-fever/). The spread of ASF has posed a devastating burden on pig industry and a huge socioeconomic impact in the worldwide. Currently, implementing strict biosecurity measures and culling of infected herds are the main ways to control its spreading since there are no safe and effective vaccines or targeted therapeutics. However, the continuous deterioration of ASF shows these control strategies to be lacking, and the lack of vaccines or therapeutic options warrants urgent further investigation (1).

ASF is caused by ASF virus (ASFV), ASFV is the only known member of the *Asfarviridae* family that belongs to the group of nucleocytoplasmic large DNA viruses. The mature ASFV particle is 260 to 300 nm in diameter and possesses a complex multilayered structure consisting of the nucleoid, core shell, inner envelope, capsid, and a host-derived outer envelope (2). The virus genome is a linear double-stranded DNA molecule of approximately 170 to 194 kbp encoding more than 150 open reading frames (ORFs). Genomic variation between strains is largely due to gain or loss of genes from the multigene families (MGFs) (3). So far, several ASFV proteins have been reported to play important roles in multiple stages of viral infection, including transcription and translation, morphogenesis, immune escape, etc. (4–6). For example, the ASFV QP509L and Q706L RNA helicases are mainly involved in viral transcription events (6, 7). The ASFV pH240R is required for the efficient production of infectious progeny virions (8). Notably, the members of MGF360 and MGF530/505 are confirmed to be IFN antagonists and play important roles in virus replication, virulence, pathogenicity, and host range (9–14).

As obligate intracellular parasites, viruses are fully reliant on host translation machinery to synthesize viral proteins. Undoubtedly, access to the translation apparatus is patrolled by intrinsic host defenses programmed to suppress viral reproduction. To ensure and optimize their own replication and spread, viruses have evolved to deploy diverse strategies to appropriate control over host translational landscape and subvert host defenses (15–17). Remarkably, a general mechanism of regulating translation initiation involves phosphorylation of the $\alpha$ subunit of eukaryotic initiation factor 2 (eIF2$\alpha$), which directs translational reprogramming and adaptation to cellular stress (18). The regulation of eIF2$\alpha$ phosphorylation has been considered a critical step for viral infection and also has an effect on virulence, tissue tropism, and pathogenicity of viruses (15–19). Consequently, in virus-infected cells, virus-host interactions that regulate eIF2$\alpha$ phosphorylation, and protein synthesis could be potential targets to develop the novel vaccines and/or antiviral therapies.

ASFV mRNAs are structurally similar to the cellular mRNAs harboring a 7-methyl-GTP cap in its 5′ untranslated region (5′UTR) and a poly(A) tail in its 3′UTR and are translated by a cap-dependent mechanism (20). Previously, it has been reported that cellular protein synthesis is strongly inhibited during ASFV infection, while viral proteins are efficiently produced (21). To subvert the function of host protein synthesis and facilitate the translation of viral mRNA in infected cells, ASFV utilizes multiple strategies, such as regulating the phosphorylation of eIF2$\alpha$, activation of eIF4E and eIF4G, recruitment of the translation machinery to viral factories, and regulation of RNA metabolism (21–23). Notably, ASFV DP71L binds host protein phosphatase 1 (PP1) to dephosphorylate eIF2$\alpha$ to avoid the shutoff of protein synthesis (24). ASFV pE66L induces the shutoff of host translation via the PKR/eIF2$\alpha$ pathway (25). ASFV ubiquitin-conjugating enzyme I215L hijacks host translation machinery to regulate the cellular protein synthesis during ASFV infection (26). However, other ASFV-encoded proteins to control the host protein synthesis machinery remain largely unknown.

ASFV MGF110 genes are the top-ranked viral genes during early and late infection (27, 28) and also linked to endoplasmic reticulum (ER) functions (29, 30), suggesting they hold importance throughout infection and provide potential candidates for antiviral or vaccine targets. In our study, we showed that MGF110-7L, a previously uncharacterized member of the MGF110 family, remarkably enhanced eIF2$\alpha$ phosphorylation via activating protein kinase R (PKR) and PKR-like ER kinase (PERK), which resulted in

host translation suppression and stress granule (SG) formation. Furthermore, proteomic analyses in combination with biochemical analyses revealed that MGF110-7L can trigger the ER stress and activate the unfolded protein response, which contributed to eIF2$\alpha$ phosphorylation and translation control. Together, this work provides the first evidence that ASFV MGF110-7L plays a role in subversion of the host cell translation machinery.

## RESULTS

**ASFV MGF110-7L functions as a potent inducer of ATF4 translation.** Phosphorylation of eIF2$\alpha$ significantly enhances the translation of activating transcription factor 4 (ATF4) mRNA by mechanisms involving two upstream ORFs (uORFs) in the 5′ leader (31). To identify ASFV proteins that induce eIF2$\alpha$ phosphorylation, we constructed expression plasmids for 179 ORFs of the ASFV China/2018/AnhuiXCGQ strain and ATF4-*Renilla* luciferase (RLuc) by replacing the main coding region of ATF4 with RLuc. We next examined their abilities to regulate translation rate of ATF4-RLuc in human embryonic kidney 293T (HEK293T) cells, which were widely used for preliminary screening of ASFV genes based on the genetic reporter systems (13, 25, 32). Thapsigargin (TG) can trigger eIF2$\alpha$ phosphorylation by inducing ER stress. In TG-treated group, expression of the ATF4-RLuc was increased ~8 times greater than basal level, whereas the ASFV DP71L, which has been previously shown to prevent the phosphorylation of eIF2$\alpha$ and the induction of ATF4 (24), was found to remarkably inhibit ATF4-RLuc expression. These findings supported the validity of our experimental approach (Fig. 1A). In these screens, most members of ASFV MGF110 family were identified to induce the translational expression of ATF4-RLuc, and MGF110-7L and MGF110-9L showed the strong stimulatory effects with >5-fold upregulation (Fig. 1A). Notably, given that other functions of MGF110-9L were reported previously (33, 34), MGF110-7L was selected for further characterization. To further examine the induction of MGF110-7L on ATF4 translation, ATF4-EGFP reporter and MGF110-7L expression plasmid were transiently cotransfected in porcine kidney-15 (PK-15) cells, the fluorescence signals were measured. In agreement with the result shown in Fig. 1A, the levels of specific fluorescence were strongly increased following TG treatment (~5.5-fold) and by the ectopic expression of MGF110-7L (~1.7-fold) (Fig. 1B). Furthermore, a Cell Counting Kit-8 (CCK-8) assay showed that ectopic expression of MGF110-7L had a negligible effect on cell viability (Fig. 1C). Collectively, these results suggest that MGF110-7L functions as a positive regulator of ATF4 translation.

**ASFV MGF110-7L induces the phosphorylation of eIF2$\alpha$ and activates the ISR.** Preferential translation of ATF4 depends on eIF2$\alpha$ phosphorylation in response to diverse stresses, and this pathway is referred to as the integrated stress response (ISR) that aims to restores balance by reprogramming gene expression (18, 35). Since the natural target cells of ASFV are macrophages, we further investigated the involvement of ASFV MGF110-7L in eIF2$\alpha$ signaling and ISR in PK-15 cells and porcine alveolar macrophage-derived 3D4/21 cells (36). Both cells were transfected with MGF110-7L-expressing plasmids with an increasing dose or empty vector and treated with TG as positive controls. As expected, we observed a significant increase in the levels of phosphorylated eIF2$\alpha$ (P-eIF2$\alpha$) and ATF4 protein in the TG-treated cells compared to untreated groups, and ectopic expression of MGF110-7L remarkably increased eIF2$\alpha$ phosphorylation and ATF4 translation in a dose-dependent manner in both cell lines (Fig. 1D and E). To further confirm the role of MGF110-7L in triggering eIF2$\alpha$ signaling and ISR activation, quantitative real-time RT-PCR (RT-qPCR) analysis was performed to evaluate the transcript levels of key ISR regulatory genes. Notably, the mRNA levels of ATF4, CHOP, and GADD34 were also significantly increased in response to increased expression of MGF110-7L (Fig. 1F to H). These results demonstrate that MGF110-7L induces the eIF2$\alpha$ signaling pathway and activates the ISR, thereby facilitating programs of stress-induced gene expression.

**ASFV MGF110-7L causes host cell translation arrest by inducing eIF2$\alpha$ phosphorylation.** The eIF2 initiation complex integrates a diverse array of stress-related signals to direct translational control and determine the translational status of global and specific mRNAs. One of the main consequences of eIF2$\alpha$ phosphorylation is the global reduction of host cell protein synthesis (Fig. 2A) (18, 37). To determine the effects of increased MGF110-7L levels on cellular translation, we addressed the global translational efficiency by measuring the

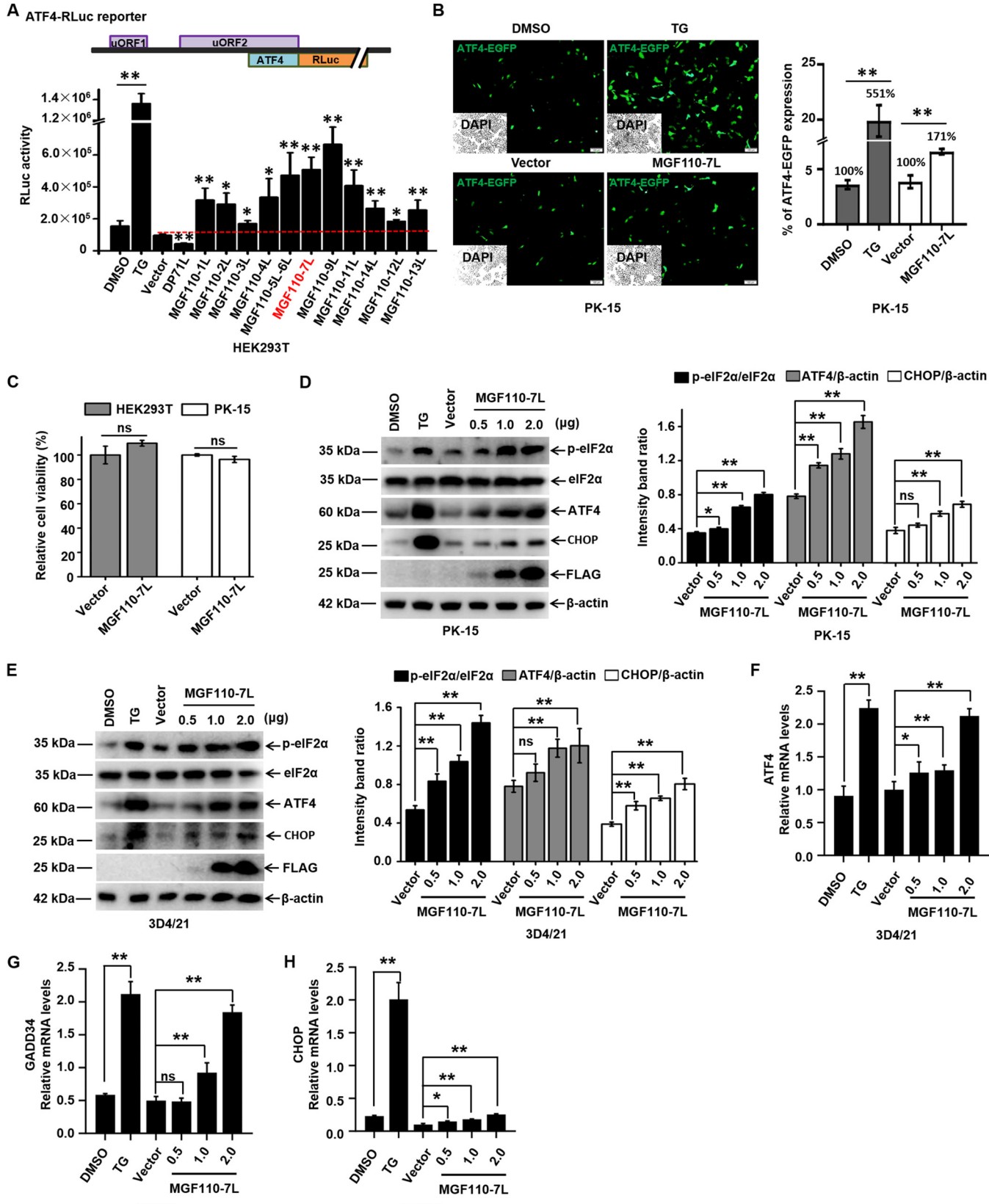

**FIG 1** ASFV MGF110-7L triggers the eIF2α signaling pathway. (A) Schematic illustration of ATF4-RLuc reporter (top). HEK293T cells were cotransfected with ATF4-RLuc reporter (0.05 μg) and an empty vector or a vector expressing different members of the MGF110 family (0.1 μg). At 24 h posttransfection, the cells were lysed, and then a luciferase assay was performed. A positive control for eIF2α phosphorylation was set by TG (1 μM) treatment for 6 h on cells

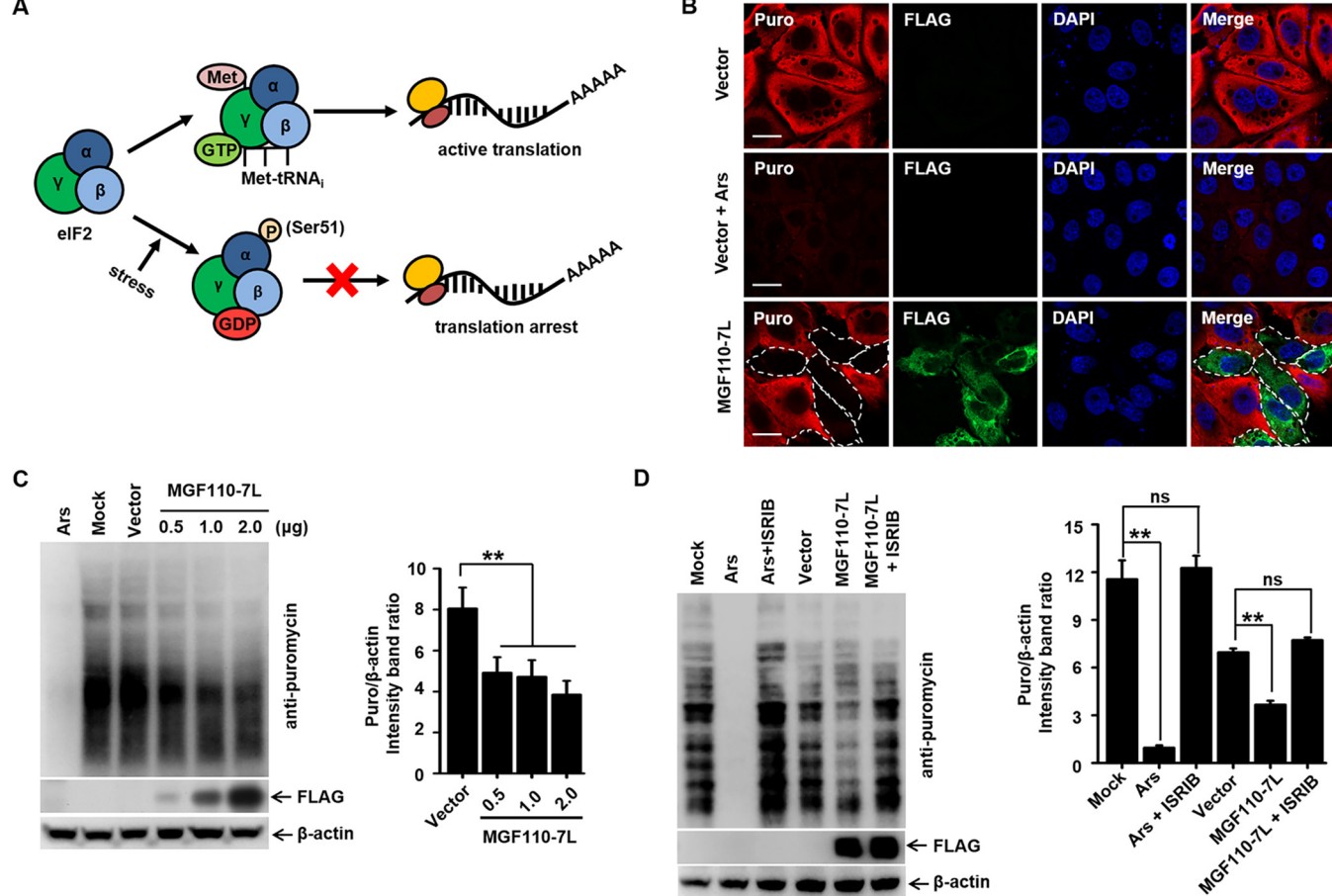

**FIG 2** Expression of ASFV MGF110-7L induces host cell protein translation shutoff. (A) Schematic diagrams showing the role of phosphorylation of eIF2$\alpha$ in host translational control. (B) PK-15 cells transfected with the FLAG empty vector or FLAG-tagged MGF110-7L (0.5 $\mu$g) for 24 h were incubated with puromycin (5 $\mu$g/mL) at 37°C for 30 min. As a positive control, cells transfected with an empty vector were treated with 0.5 mM arsenite (Ars) for 45 min before addition of puromycin. The cells were then fixed, permeabilized, and processed for IFA. A FLAG-specific monoclonal antibody was used to detect MGF110-7L (green). Puromycylated chains are visualized using an antipuromycin antibody (red). Nuclei were stained with DAPI (blue). Scale bar, 20 $\mu$m. (C) 3D4/21 cells were transfected with the MGF110-7L-Flag-expressing vector with an increasing dose (0.5, 1.0, and 2.0 $\mu$g) or an empty Flag vector (2.0 $\mu$g) for 24 h, at which point the cells were pulsed with puromycin as described above, and whole-cell lysates were obtained and immunolabeled with anti-Flag and anti-puromycin antibodies (left). Densitometry analysis of puromycin signal in MGF110-7L-transfected cells compared to empty vector-transfected cells was performed by ImageJ (right). (D) 3D4/21 cells were treated with 0.5 mM arsenite (Ars) for 45 min, or transfected with the FLAG empty vector or FLAG-tagged MGF110-7L (2.0 $\mu$g) for 24 h, with (+ISRIB) or without (−ISRIB) 0.5 $\mu$M ISRIB. At 30 min before the cell lysate samples were harvested, the cells were incubated with puromycin and immunolabeled with anti-puromycin antibodies to visualize the nascent polypeptidic chains. The data are means ± the SD of results of three independent experiments. **, $P < 0.01$; ns, not significant.

incorporation of puromycin by confocal microscopy and Western blot analysis (38). As a positive control, treatment with sodium arsenite, a potent inducer of eIF2$\alpha$ phosphorylation (39), resulted in a profound absence of puromycin incorporation in PK-15 cells and 3D4/21 cells. Intriguingly, we also observed a significant and dose-dependent decrease in global protein synthesis in MGF110-7L-expressing cells (Fig. 2B and C).

Due to our observations that MGF110-7L-induced eIF2$\alpha$ phosphorylation status corresponds to global host translation arrest, we hypothesized that inhibition of the phosphorylation

**FIG 1** Legend (Continued)

transfected with ATF4-RLuc. (B) PK-15 cells were cotransfected with ATF4-EGFP reporter (0.2 $\mu$g) and FLAG vector or MGF110-7L-Flag (0.4 $\mu$g). At 24 h posttransfection, the cells were subjected to fluorescence analysis. TG was used as the positive control. Quantification of ATF4-EGFP expression was done using ImageJ software in at least 10 random fields of view with greater than 800 cells analyzed on each slide. Bar graphs on the right show the percentage of cells expressing EGFP in each group under different treatments. (C) HEK293T and PK-15 cells were transfected with an empty vector or MGF110-7L-FLAG plasmid (2 $\mu$g) for 24 h. Cell viability was assessed with CCK-8 assays. (D and E) PK-15 and 3D4/21 cells were transfected with a MGF110-7L-Flag-expressing vector with an increasing dose (0.5, 1.0, 2.0 $\mu$g) or an empty Flag vector (2.0 $\mu$g) for 24 h. TG was used as a positive control. The cells were then subjected to Western blot analysis (left). The grayscale values of the protein bands were analyzed by ImageJ (right). (F to H) 3D4/21 cells were treated as described above for panel E. RNA samples were extracted at the indicated times, and the mRNA levels of ATF4 (F), GADD34 (G), and CHOP (H) were determined by RT-qPCR analysis. The data are the means of results of three independent experiments, and error bars indicate the SD. *, $P < 0.05$; **, $P < 0.01$; ns, not significant.

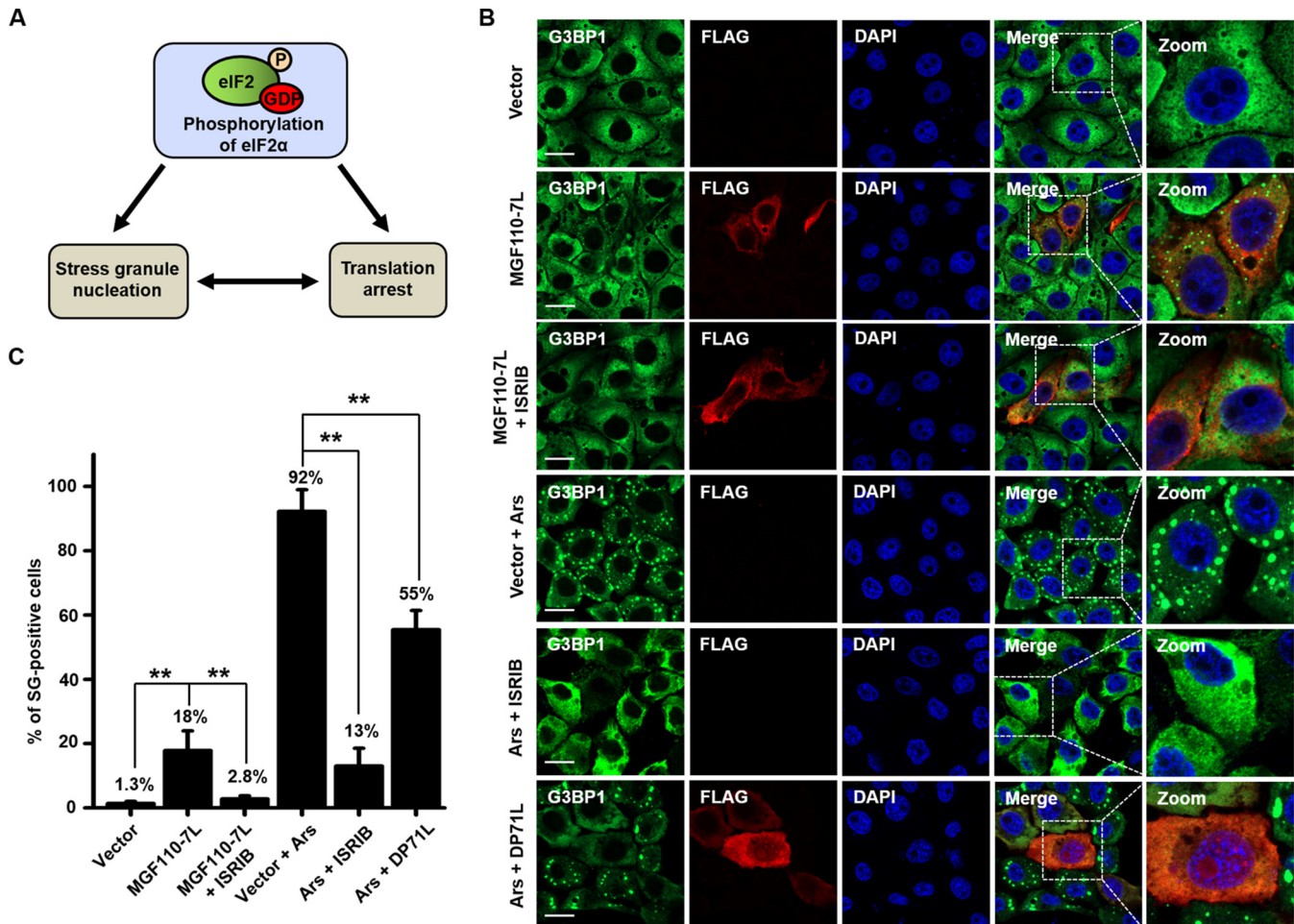

**FIG 3** Expression of ASFV MGF110-7L induces SG formation. (A) Schematic diagram showing the relationship of eIF2α phosphorylation, translation arrest, and SG formation. (B) PK-15 cells were transfected with an empty Flag vector or a MGF110-7L-Flag-expressing vector (0.5 μg) for 24 h, with or without 0.5 μM ISRIB. As a control, cells transfected with FLAG vector or DP71L-FLAG (0.5 μg) for 24 h were treated with 0.5 mM arsenite (Ars) for 45 min, with or without 0.5 μM ISRIB. The cells were then fixed, permeabilized, and incubated with anti-G3BP1 and anti-FLAG antibodies and then with secondary antibodies conjugated with AF488 (green) and AF594 (red), respectively. Nuclei were counterstained with DAPI (blue). Cells were observed by confocal microscopy. Scale bar, 20 μm. (C) Representative bar plots show the percentage of cells with SGs counted from at least 100 cells in each group under different stress conditions. The data are means ± the SD from three independent experiments. **, $P < 0.01$.

level of eIF2α would restore the reduction of protein synthesis. After transfection of MGF110-7L expression vector and treatment with ISRIB, a known pharmacologic ISR inhibitor that has previously been shown to reverse the inhibitory impact of P-eIF2α (40), 3D4/21 cells were pulsed with puromycin for 30 min before harvesting lysates at 24 h postinfection. Western blot analysis showed that ISRIB treatment of MGF110-7L-expressing cells significantly restored the global translational efficiency compared to untreated (Fig. 2D). Taken together, these results strongly suggested that MGF110-7L-induced global translation arrest is eIF2α phosphorylation dependent.

**ASFV MGF110-7L triggers SG formation in an eIF2α phosphorylation-dependent manner.** Stalling of translation initiation by increased P-eIF2α leads to polysome disassembly triggering SG formation, which are large cytoplasmic foci that are nucleated by the aggregation of untranslated mRNAs and associated 40S ribosomal subunits and proteins. Consequently, SGs can serve as specialized signaling platforms to intensify stress-induced translation arrest and form a unique clustering interface (Fig. 3A) (18, 41). Accordingly, to determine whether MGF110-7L-induced translation repression was linked to SG assembly, PK-15 cells were transfected with a MGF110-7L-expressing or empty vector or treated with arsenite, and SG assembly was then monitored by indirect immunofluorescence assay (IFA). Ras-GAP SH3 domain-binding protein 1 (G3BP1) was used as a *bona fide* SG marker (42). As

shown in Fig. 3B and C, we observed that almost no SGs (with only 1.3% of cells) were observed in empty vector-transfected cells, and arsenite induced the formation of cytoplasmic puncta in 92% of cells. Intriguingly, ectopic expression of MGF110-7L also resulted in the accumulation of G3BP1 in large cytoplasmic foci in around 18% of cells. Next, to further determine whether MGF110-7L-induced SG formation was correlated with the P-eIF2$\alpha$ signaling, we assessed the response to arsenite- or MGF110-7L-induced stress upon ISRIB treatment. As expected, ISRIB was found to result in a dramatic reduction of arsenite-induced SGs, with ~13% of cells containing SGs. Moreover, the ASFV DP71L, an eIF2$\alpha$ phosphorylation antagonists, also blocked arsenite-induced SG assembly. These findings indicated that the phosphorylation status of eIF2$\alpha$ is positively correlated with SG formation. Importantly, ISRIB treatment of MGF110-7L-expressing cells also substantially decreased the numbers of SGs compared to untreated cells, with only 2.8% of cells containing SGs (Fig. 3B and C). Altogether, these results reveal that MGF110-7L induces SG formation and the process is dependent upon the phosphorylation of eIF2$\alpha$.

**ASFV MGF110-7L induces eIF2$\alpha$ phosphorylation by activating the PERK and PKR pathways.** During cellular stress, eIF2$\alpha$ can be phosphorylated by any of four cellular kinases, PERK, PKR, GCN2 (general control nonderepressible 2), and HRI (heme-regulated inhibitor) (Fig. 4A) (37). To define the mechanism of MGF110-7L-mediated induction of eIF2$\alpha$ phosphorylation, we investigated the activation and phosphorylation of the eIF2$\alpha$ kinases, PERK, PKR, and GCN2. As expected, stimulation of 3D4/21 and PK-15 cells with TG induced PERK phosphorylation, but not PKR or GCN2 (Fig. 4B and C). Strikingly, we observed that a dose dependent, increased expression of MGF110-7L exhibited a progressive increase in PERK and PKR phosphorylation but had no effect on the phosphorylation of GCN2 (Fig. 4B and C). To further verify the role of PERK and/or PKR in MGF110-7L-mediated induction of eIF2$\alpha$ phosphorylation, we utilized a PERK inhibitor (GSK2606414) and a PKR inhibitor (C16) to treat MGF110-7L-expressing or control cells. First, cell viability and the inhibitory effect of GSK2606414 and C16 were determined, and the results showed that PERK and PKR-mediated phosphorylation of eIF2$\alpha$ was suppressed by specific inhibitor (Fig. 4D to F). Moreover, treatment of cells with GSK2606414 or C16 substantially decreased MGF110-7L-induced phosphorylation of PERK, PKR, and eIF2$\alpha$ (Fig. 4E and F). Altogether, these results reveal that MGF110-7L-induced phosphorylation of eIF2$\alpha$ is primarily mediated via PERK and PKR.

**Physical and biochemical parameters of ASFV MGF110-7L.** To explore the underlying mechanism of MGF110-7L-mediated activation of the PERK/PKR-eIF2$\alpha$ pathways, we next analyzed its features of sequence, structure, and function. To assess the conservation of MGF110-7L, we selected 12 representative sequences from Asia, African, and European pathogenic virus strains and aligned them using ClustalX 2.1 software. The alignment analysis showed that MGF110-7L is highly conserved among the strains and possesses a highly conserved cysteine-rich motifs center and a C-terminal ER retention signal KKDEF (Fig. 5A), suggesting that MGF110-7L can form disulfide bonds and function in an oxidizing environment in the ER lumen. MGF110-7L was also predicted to have a mixed composition of secondary structures containing three helices and six sheets. The secondary structure exhibits high hydropathicity containing the signal peptide within the N-terminal 20 amino acids and N-linked glycosylation site at positions Asn70 and Asn105, whereas no transmembrane helix (Fig. 5A and B). In addition, we also analyzed the tertiary structure and the underlying ligand binding sites of MGF110-7L using I-TASSER, and identified a cavity by residues Cys62 and Cys92 for ligand binding (Fig. 5C).

**ASFV MGF110-7L is primarily located in the ER.** Based on the potential signal peptides, MGF110-7L may involve in the secretory pathway, and the process of protein sorting directs MGF110-7L to a specific destination in cells. In order to define the subcellular localization of MGF110-7L, MGF110-7L-expressing vector, together with one of the organelle-specific markers (which encodes a fusion of DsRed2 and the targeting sequence of the corresponding organelle), was cotransfected to PK-15 cells. As shown in Fig. 6, MGF110-7L was diffusely distributed in the cells and colocalized predominantly with the ER marker, whereas a partial colocalization was observed between MGF110-7L and the marker of Golgi apparatus, mitochondria, lysosome, or peroxisome. Calculating of the overlap coefficient (R) and colocalization profile between MGF110-7L and organelle markers further confirmed the

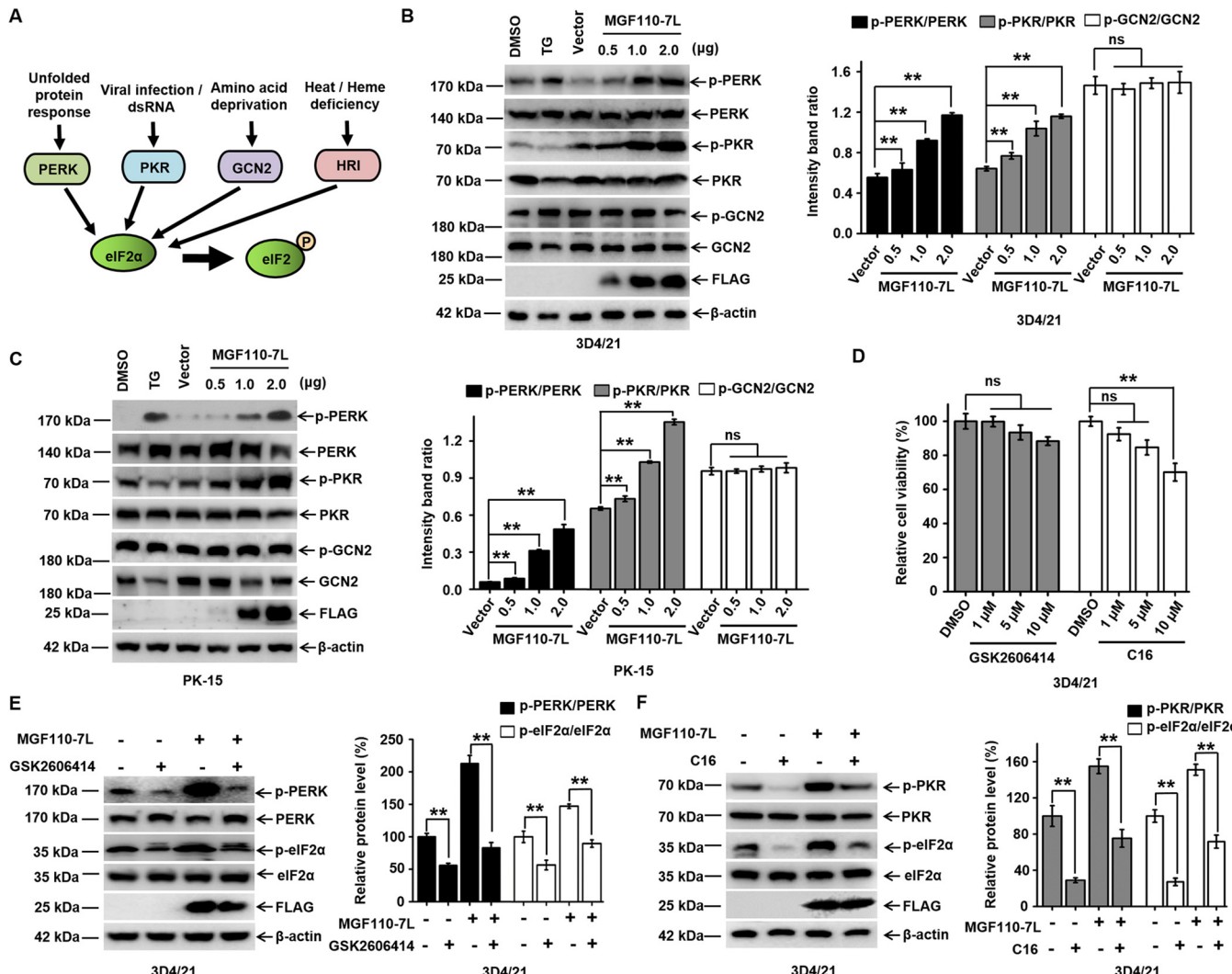

**FIG 4** Expression of ASFV MGF110-7L enhances eIF2α phosphorylation through upregulation of PERK and PKR activity. (A) Diagrams of four major stress pathways leading to eIF2α phosphorylation by different kinases, PERK, PKR, GCN2, and HRI. (B and C) 3D4/21 and PK-15 cells were transfected with the MGF110-7L-Flag-expressing vector with an increasing dose (0.5, 1.0, 2.0 μg) or an empty Flag vector (2.0 μg) for 24 h. As a positive control, the cells were treated with TG (1 μM) for 6 h. The whole-cell lysates were then obtained and immunolabeled using the indicated antibodies (left). Densitometry analysis of these protein bands in MGF110-7L-transfected cells compared to empty vector-transfected cells was performed using ImageJ (right). (D) 3D4/21 cells were treated with the indicated concentrations (1, 5, and 10 μM) of GSK2606414 and C16 for 24 h, and cell viability was measured using a CCK-8 assay. (E and F) 3D4/21 cells were transfected with an empty Flag vector or a MGF110-7L-Flag-expressing vector (2.0 μg), with or without GSK2606414 (10 μM) or C16 (1 μM) as indicated for 24 h before cell lysate samples were obtained. Lysates were analyzed via immunoblotting with the indicated antibodies. The relative levels of p-PERK, p-PKR, or p-eIF2α in each sample after normalizing to the corresponding total PERK, total PKR, or total p-eIF2α was determined using ImageJ software and plotted in bar graphs. Data in panels B to D show means ± the SD of the results of three independent experiments. **, $P < 0.01$, ns, not significant.

above results (Fig. 6). Moreover, the fluorescence confocal also showed that, in the control cell, pDsRed2-Golgi displayed a bow-like pattern of fluorescence in the perinuclear area and pDsRed2-Peroxi exhibited punctate-like structures with diffuse distribution throughout the cytoplasm. Strikingly, ectopic expression of MGF110-7L triggered a significant reorganization of the subcellular distribution and morphological characteristics of the *trans*-Golgi network and peroxisome (Fig. 6). Taken together, these results indicated that MGF110-7L is predominantly located in the ER and a small amount is retained in other intracellular compartments within the secretory pathway, along with disruption of the structural components of specific organelles, suggesting the involvement of MGF 110-7L in the secretory pathway.

**Identification of ASFV MGF110-7L-interacting host factors involved in modulation of ER redox homeostasis.** To further explore the potential mechanism of MGF110-7L-induced translation arrest and SG formation in the downstream of eIF2α phosphorylation, we performed the high-throughput proteomics analysis by using a combination of

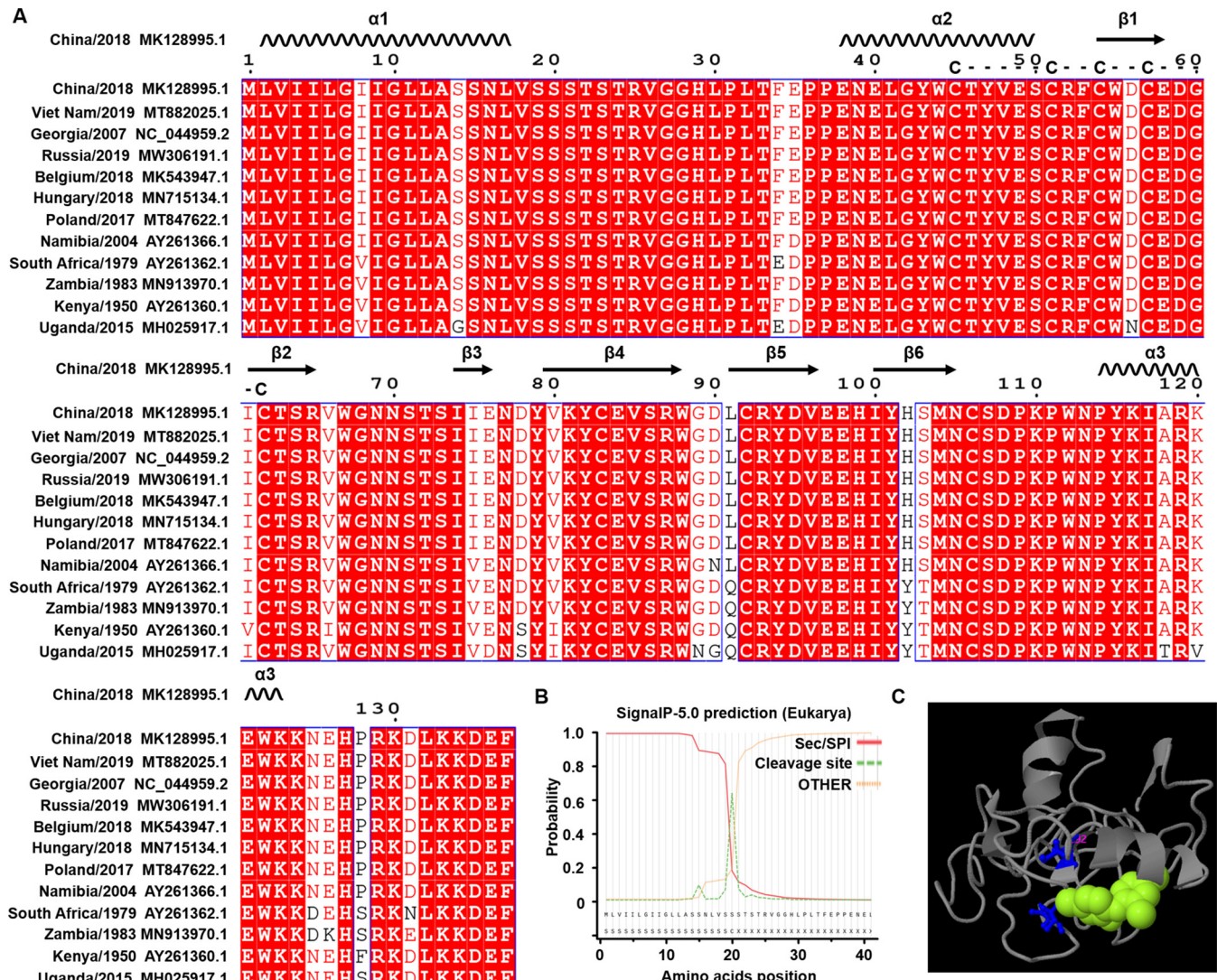

**FIG 5** Sequence alignment, structure, and function analysis of ASFV MGF110-7L protein. (A) Multiple sequence alignment of the indicated ASFV isolates of viral protein MGF110-7L were performed using ClustalX 2.1, and the figure was prepared with ESPript3.0. Residues conserved in all sequences are shown in white on a red background. The predicted secondary structures and conserved cysteine-rich domains of MGF110-7L are indicated above the sequences. (B) Predicted signal peptide and cleavage sites of MGF110-7L. (C) Visualization of the tertiary structure and the underlying ligand binding sites of MGF110-7L.

immunoprecipitation (IP) and liquid chromatography–tandem mass spectrometry (LC-MS/MS) to understand the interaction between MGF110-7L and host proteins in HEK293T cells (Fig. 7A). We carried out the experiments with two independent biological replicates. The overview of the identified 81 cellular proteins interacting with MGF110-7L from two quantitative mass spectrometry analysis was presented in Table S1 in the supplemental material. Subsequently, the identified proteins were searched against the GO database to obtain enrichment information of biological process, cellular component, and molecular function. GO analysis showed (i) that in the biological process of the identified proteins, the proteins were mainly involved in oxidation-reduction and proteolysis involved in cellular protein catabolic and and metabolic processes; (ii) that in the cellular component, the top three most significant enrichment patterns were the integral component of membrane, the membrane, and the proteasome core complex; and (iii) that in molecular functions, the proteins were involved in protein binding, ATP binding, and oxidoreductase activity (Fig. 7B). Next, STRING network analysis was performed to elucidate the known and predicted associations between MGF110-7L interacting host proteins, and 611 interactions were observed at a 0.4 confidence level. A few high local network clusters were found in the group of proteins involved in the proteasome, spliceosome, ER membrane protein complex, and calcium signaling pathway

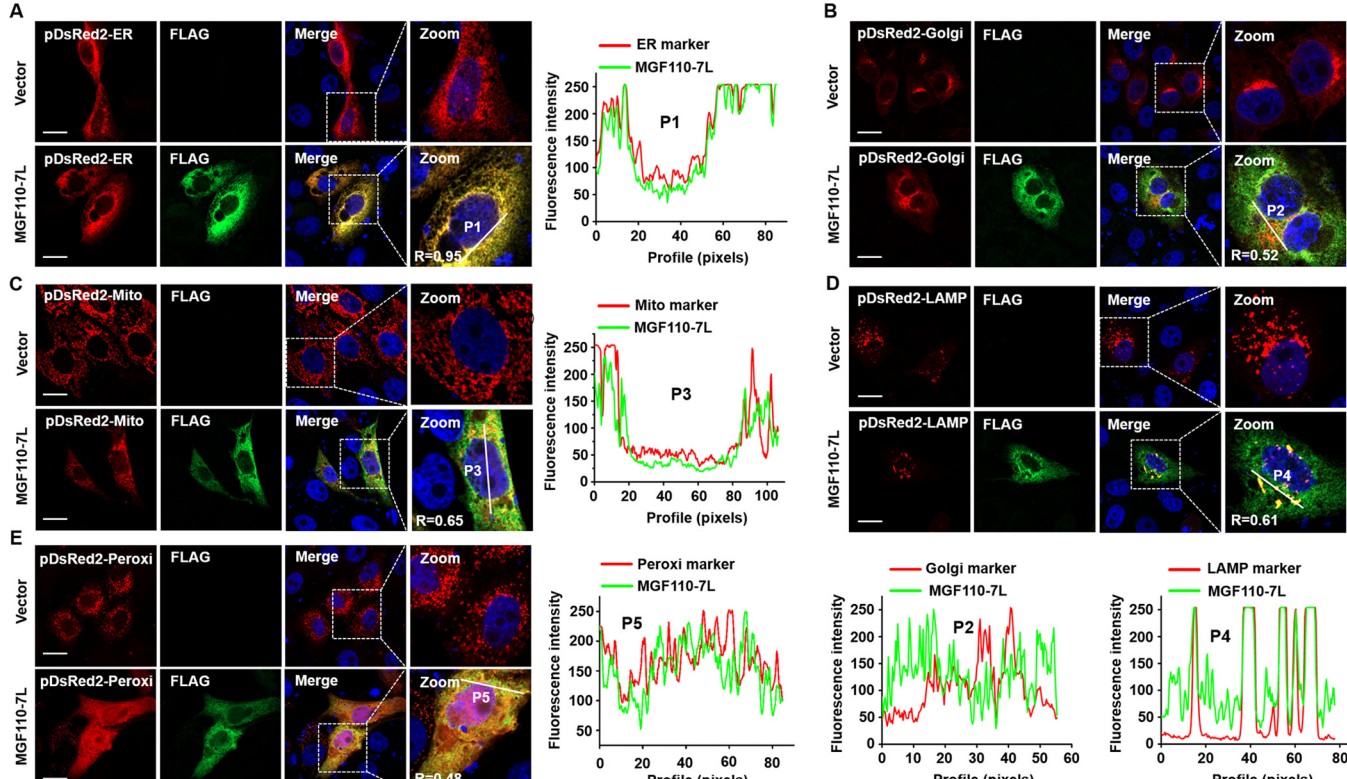

**FIG 6** Confocal immunofluorescence analysis of the intracellular distribution of ASFV MGF110-7L protein. PK-15 cells were cotransfected with an empty Flag vector or a MGF110-7L-Flag-expressing vector (0.5 $\mu$g) and one of the following organelle markers: pDsRed2-ER (A), pDsRed2-Golgi (B), pDsRed2-Mito (C), pDsRed2-LAMP1 (D), or pDsRed2-Peroxi (E) (0.2 $\mu$g). At 24 h posttransfection, the cells were fixed, permeabilized, and incubated first with anti-FLAG antibody and then with secondary antibody conjugated with AF488 (green). Nuclei were counterstained with DAPI (blue). The organelle marker was directly visualized (red), and localization was determined using confocal microscopy. The overlapping coefficient (R) was shown in enlarged images, and the intensity profile of the linear region of interest (ROI) across the PK-15 cell contained with MGF110-7L and the indicated organelle markers. Scale bar, 20 $\mu$m.

(Fig. 7C). Further analysis of these interactions by KEGG pathway enrichment identified the 10 major cellular pathways. Interestingly, these tight interactions of proteins were mainly enriched in the proteasome, spliceosome, protein processing in ER, and metabolic pathways (Fig. 7D). Based on the proteomic analysis and the subcellular localization, it was speculated that the cellular proteins interacting with MGF110-7L are mainly involved in ER redox homeostasis, posttranslational protein modification, the proteasome pathway, and the cellular response to stress.

To verify the protein interactions and to determine whether MGF110-7L is involved in ER stress in the ASFV early-replication stage (27), co-IP experiments were performed *in vitro*. Three cellular proteins were selected from MGF110-7L-interacting cellular proteins, such as transmembrane emp24 domain-containing protein 4 (TMED4), protein disulfide-isomerase A3 (PDIA3), and proteasome subunit alpha type-4 (PSMA4), which were all involved in the regulation of ER homeostasis (Fig. 8). We found MGF110-7L interacted with PDIA3, TMED4, and PSMA4 (Fig. 8A to C). Meanwhile, double-label immunofluorescence experiments showed that MGF110-7L could colocalize with PDIA3, TMED4, and PSMA4, which supports the co-IP analysis (Fig. 8D to F). Overall, these data verified the validity of our observations on proteomic analysis and provided a comprehensive spectrum of host proteins that might be involved in MGF110-7L-induced cellular stress.

**ASFV MGF110-7L induces ER stress and activates the unfolded protein response.** To further determine whether MGF110-7L alone is sufficient to induce ER stress, we investigated the expression level of the major ER stress marker BIP after transfection of MGF110-7L-expression plasmids in 3D4/21 cells. Using immunoblotting analysis, we observed that treatment with TG resulted in elevated expression levels of the chaperone BIP compared to the untreated control, and the expression of MGF110-7L remarkably triggered the upregulation of BIP protein

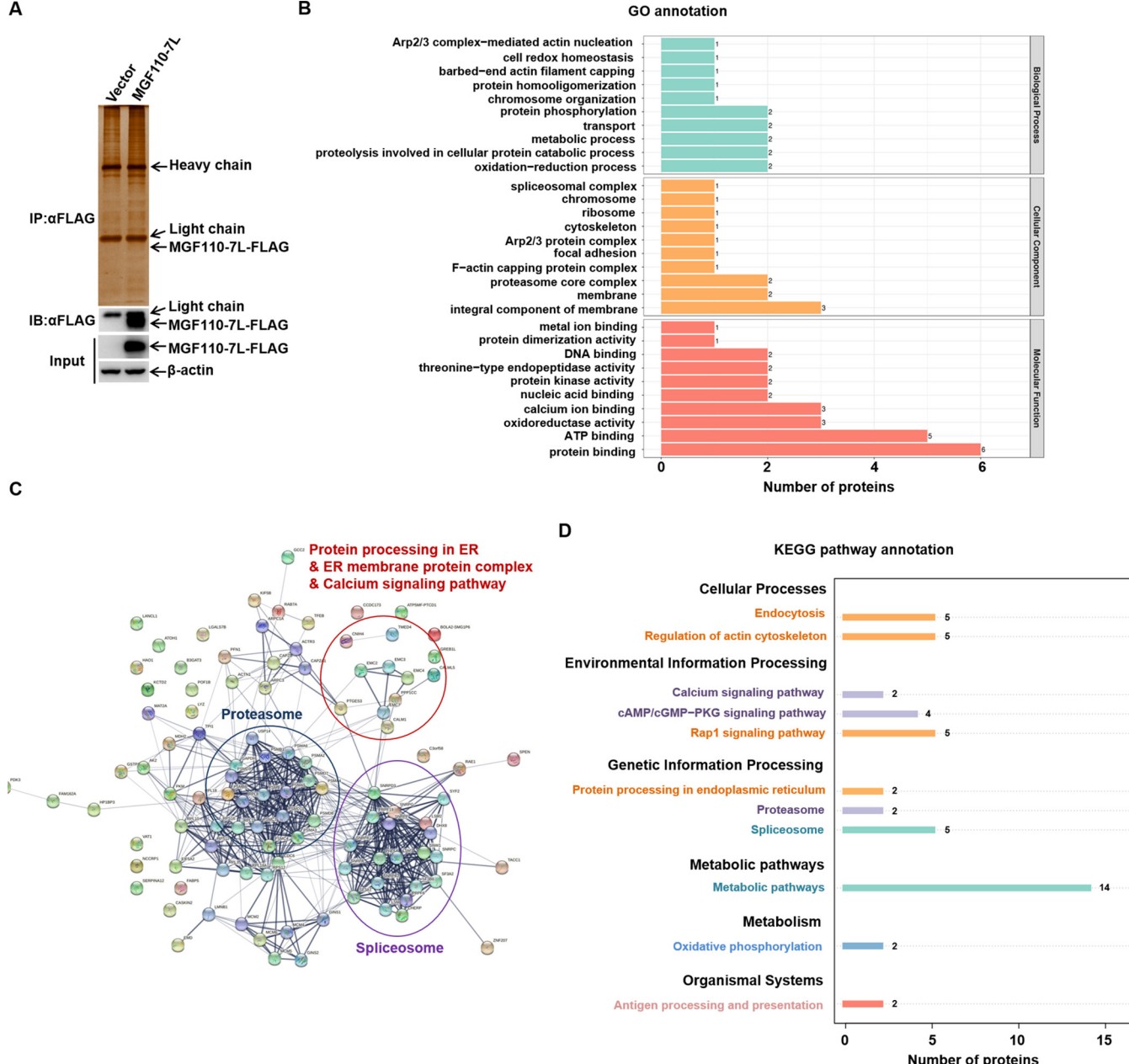

**FIG 7** Proteomics analysis of ASFV MGF110-7L-host interactome. (A) Western blot and silver staining were used to detect the expression of exogenous MGF110-7L in HEK293T cells and the enrichment of MGF110-7L-interacting proteins. The data were tested two times independently. (B) GO enrichment analysis of all the cellular proteins interacting with MGF110-7L. Based on the annotations from GO, these cellular proteins were categorized into different biological processes, cellular components, and molecular functions. (C) Network interactions of the MGF110-7L-interacting proteins by STRING analysis at confidence level 0.4. The interactions between these cellular proteins are indicated by the connecting line. The thickness of the connecting line represents the strength of the associations. Proteins functionally categorized under some enriched groups based on existing information from Swiss-Prot/TrEMBL database are highlighted. (D) KEGG pathway enrichment analysis. The enriched pathways targeted by MGF110-7L-interacting proteins were analyzed using the KEGG functional annotation pathway database.

cells in a dose-dependent manner (Fig. 9A). As expected, the mRNA levels of BIP were also gradually increased following a dose-dependent expression of MGF110-7L (Fig. 9B). These data confirmed that MGF110-7L expression can trigger ER stress.

To relieve ER stress, three ER transmembrane sensors (PERK, IRE1, and ATF6) of the unfolded protein response (UPR) pathway may be activated (43, 44). Our results demonstrated that the ectopic expression of MGF110-7L activated the PERK-eIF2$\alpha$-ATF4 pathway (Fig. 1). To further assess the status of the IRE1$\alpha$-XBP1 and ATF6 branches of the UPR following MGF110-7L expression, we analyzed the levels of phosphorylated IRE1$\alpha$, 90-kDa full-length ATF6

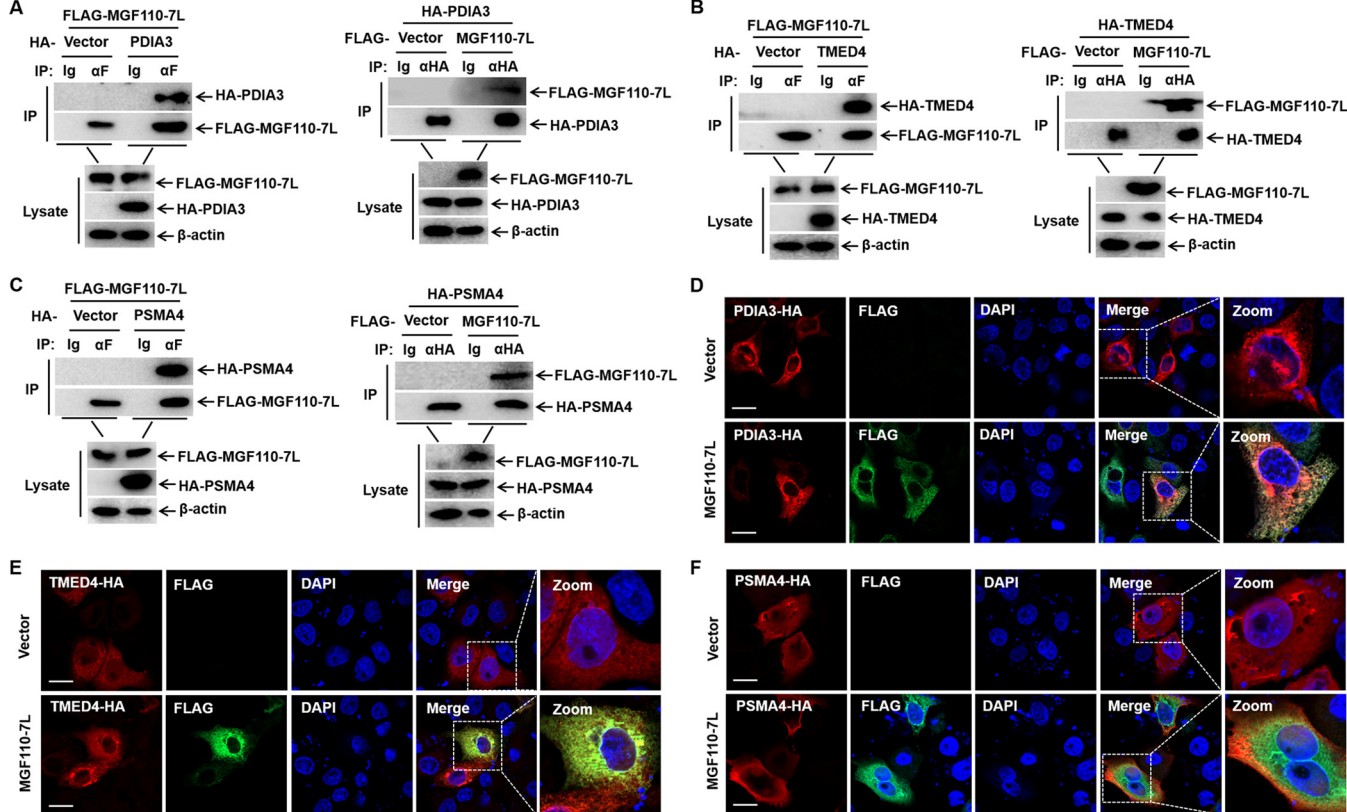

**FIG 8** Validating the interactions between cellular proteins with ASFV MGF110-7L protein. (A to C) HEK293T cells were cotransfected with an empty Flag vector or a MGF110-7L-Flag-expressing vector (5 μg) and one of the following plasmids expressing PDIA3-HA (A), TMED4-HA (B), or PSMA4-HA (C) (5 μg). At 24 h posttransfection, the cells were lysed and immunoprecipitated with anti-HA or anti-Flag antibodies, followed by Western blotting. (D to F) PK-15 cells were cotransfected with an empty Flag vector or a MGF110-7L-Flag-expressing vector (0.5 μg) and one of the following plasmids expressing PDIA3-HA (D), TMED4-HA (E), or PSMA4-HA (F) (0.5 μg). At 24 h posttransfection, the cells were fixed and permeabilized for IFA. Cells were immunolabeled with antibodies against FLAG (green), HA (red), and DAPI (blue). Samples were captured by confocal microscope. Scale bar, 20 μm.

(ATF6p90), and 50-kDa active variant of ATF6 (ATF6p50) in 3D4/21 cells. Using immunoblotting analysis, we found that the expression of MGF110-7L significantly enhanced the phosphorylation level of IRE1α in a dose-dependent manner but had no effect on the level of activated ATF6p50. To further validate the above observations, the mRNA levels of XBP1(s) and ATF6p50-targeted UPR genes were detected by RT-qPCR. The mRNA levels of processed Xbp1 mRNA [XBP1(s)/XBP1(u)], EDEM1, P58IPK, and ERdj4 were significantly elevated in MGF110-7L-expressing cells (Fig. 9C to F). Nevertheless, the expression of MGF110-7L did not result in significant induction of GRP94 and Calreticulin, two transcriptionally induced downstream genes of ATF6p50 (Fig. 9G and H). Consistent with the RT-qPCR results, strong nuclear signals of ATF6 were observed in TG-treated PK-15 cells, but not in MGF110-7L-expressing cells (Fig. 9I). Taken together, these results demonstrate that MGF110-7L triggers ER stress and activates the PERK and IRE1α-XBP1 branches of the UPR to different extents, but not the ATF6-mediated UPR pathway.

## DISCUSSION

Despite exhibiting an incredible degree of self-sufficiency and complexity (5, 6), ASFV remains unconditionally dependent on the host translation machinery to synthesize viral proteins. The findings reported here reveal a previously uncharacterized role of MGF110-7L in subversion of the host cell translation machinery and provide insights into translational control during ASFV infection. Importantly, we examined how ASFV MGF110-7L interacts with the eIF2α signaling axis controlling translational reprogramming, and we addressed the role of MGF110-7L in induction of the cellular stress responses, eIF2α phosphorylation, translation inhibition, and SG formation (Fig. 10).

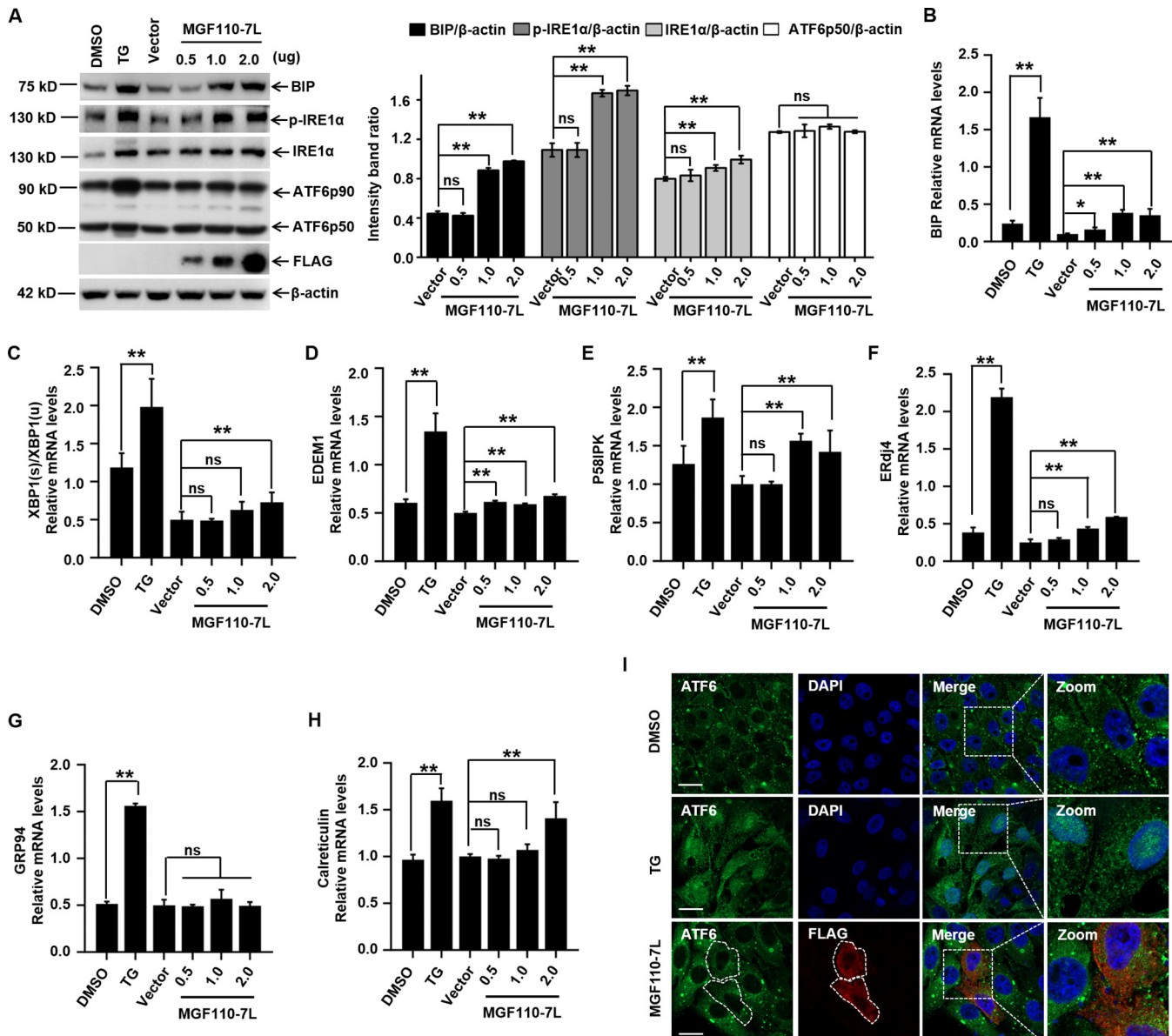

**FIG 9** ASFV MGF110-7L induces ER stress. (A) 3D4/21 cells were transfected with a MGF110-7L-Flag-expressing vector with an increasing dose (0.5, 1.0, and 2.0 $\mu$g) or an empty Flag vector (2.0 $\mu$g) for 24 h. A positive control for ER stress was set by TG (1 $\mu$M) treatment for 6 h. Whole-cell lysates were obtained and immunolabeled with the indicated antibodies. Densitometry analysis of these protein bands in MGF110-7L-transfected cells normalized to empty vector-transfected cells was performed by ImageJ (right). (B to H) 3D4/21 cells were treated as described above for panel A. RNA samples were extracted at the indicated times, and the mRNA levels of BIP (B), XBP1(s)/XBP1(u) (C), EDEM1 (D), P58IPK (E), ERdj4 (F), GRP94 (G), and Calreticulin (H) were determined by RT-qPCR analysis. (I) Representative immunofluorescence images of PK-15 cells transfected with MGF110-7L-Flag-expressing vector (0.5 $\mu$g) for 24 h or treated with TG (1 $\mu$M) for 6 h and then incubated with anti-ATF6 (green) and anti-FLAG (red) antibodies. Nuclei were counterstained with DAPI (blue). Scale bar, 20 $\mu$m. Data in panels A to H show means $\pm$ the SD of the results of three independent experiments. *, $P < 0.05$; **, $P < 0.01$; ns, not significant.

eIF2$\alpha$ phosphorylation is one of the most important host defense mechanisms against viral infections, and eIF2$\alpha$ signaling could be a superior target for viruses to dominate the infected cell translational landscape (15–19). Notably, it has been reported that eIF2$\alpha$ remained unphosphorylated throughout the productive infection of cell-adapted strain ASFV-Ba71V infection in Vero cells (21), while in murine Raw cells, the sustained eIF2$\alpha$ phosphorylation was observed after the virulent isolate E70 addition, concomitant with the blockage of the infection (22). These features proposed that targeting eIF2$\alpha$ phosphorylation may contribute to the efficient replication of ASFV. To uncover how ASFV manipulates the eIF2$\alpha$ signaling, we systematically screened the effects of all of ASFV-encoded proteins on the P-eIF2$\alpha$ levels and identified several MGF110 members, including MGF110-9L, MGF110-7L, and MGF110-5L-6L, had a

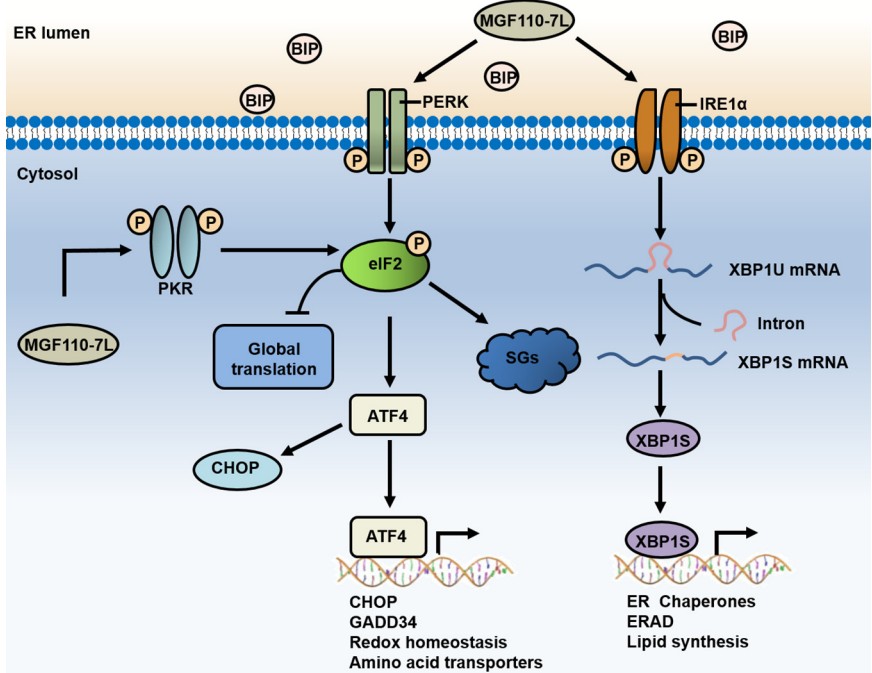

**FIG 10** Schematic representation of the proposed role of ASFV MGF110-7L in subversion of the host protein translation. ASFV-encoded MGF110-7L could trigger ER stress and activate the PERK and IRE1$\alpha$-XBP1 branches of the UPR. The ER stress and UPR induction promotes the phosphorylation of eIF2$\alpha$ by activating the kinases PERK and PKR, resulting in host translation arrest and SG formation.

significant regulatory effect. It is noteworthy that the MGF110-7L gene was recently shown to be the top-ranked viral gene at any time point during infection (27, 28), suggesting its high importance for viral reproduction.

In this study, our results indicated that exogenous expression of MGF110-7L in both 3D4/21 and PK-15 cells both could induce ER stress and activate the PERK and IRE1$\alpha$-XBP1 branches of the UPR pathway, which contributes to the phosphorylation of eIF2$\alpha$ by activating the kinases PERK and PKR. Consequently, MGF110-7L-induced phosphorylation of eIF2$\alpha$ results in host cell translation arrest and SG formation (Fig. 10). Theoretically, stress responses leading to cell translational reprogramming and SG formation may be detrimental to virus replication. However, ASFV is likely to employ remarkable strategies to manipulate eIF2$\alpha$-mediated translational control and counter the hostile environment to favor the productive infection. ASFV early protein pE66L is an ER transmembrane protein, and it has been shown that pE66L significantly suppresses host protein synthesis by activating the PKR/eIF2$\alpha$ pathway, whereas depletion of the E66L gene had negligible effect on virus replication (25). The late viral protein DP71L has been shown to inhibit the global shutoff of protein synthesis by recruiting host PP1 to dephosphorylate eIF2$\alpha$, but depletion of this gene from ASFV did not increase the phosphorylation of eIF2$\alpha$. These data imply that the virus may have other complementary functions to avoid eIF2$\alpha$ phosphorylation (24). In agreement with previous reports, our screening analyses further identified that most of ASFV-encoded early genes significantly stimulate eIF2$\alpha$ phosphorylation, while the late genes reduce eIF2$\alpha$ phosphorylation (data not shown). Consequently, it is tempting to speculate that, as ASFV infection progressed, viral early proteins are likely responsible for enhancing P-eIF2$\alpha$ levels to arrest host translation and counter host defenses, and specific late proteins may be involved in inhibition of eIF2$\alpha$ phosphorylation to promote viral proteins synthesis.

Previous studies have demonstrated that ER as a site for ASFV replication and maturation, and these processes can trigger ER stress and activate the UPR (45, 46). Furthermore, a recent study showed that ASFV infection induces the upregulation of PERK-mediated UPR pathway in the early-replication stage (27). In order to determine whether early viral protein MGF110-7L is involved in this process and explore its underlying mechanisms, we analyzed

the structure, subcellular localization, and interaction network of MGF110-7L with proteins. Similar to most members in the MGF110 family, MGF110-7L was predicted to contain the signal peptide, a highly conserved central cysteine-rich domains, and a C-terminal ER retention signal (30, 47). Using confocal immunofluorescence technique, MGF110-7L was confirmed to be mainly located in the ER lumen and that a small amount is retained in other intracellular organelles, including the Golgi apparatus, mitochondria, lysosomes, and peroxisomes. MGF110-7L also triggers a significant reorganization of the subcellular distribution and morphological characteristics of the Golgi apparatus and peroxisome, suggesting that this protein plays important roles in the process of remodeling ER/Golgi apparatus, protein sorting at the ER-Golgi apparatus interface, and peroxisome generation. Our subsequent proteomic analysis revealed that the cellular proteins interacting with MGF110-7L are mainly involved in protein processing in ER, posttranslational protein modification, proteasome pathway, and cellular response to stress. Combined with biochemical experiments, it is speculated that MGF110-7L may disrupt the cellular physiological homeostasis by affecting the redox homeostasis in the ER lumen, the protein secretory pathway, or integral components of membrane organelles and hence induce a series of stress responses such as ER stress. Remarkably, ASFV infection disrupts the *trans*-Golgi network, which is linked to a general reorganization of the secretory pathway, and ultimately prevents the delivery of host immunoregulatory MHC-I proteins to the plasma membranes of macrophages (48, 49). It is conceivable that these strategies employed by ASFV may not only block these effects that are detrimental to its infection but also benefit from those effects to generate favorable conditions for virus reproduction, like serving for viral morphogenesis, formation of viral factories, and evasion of host immune response. The detailed mechanisms are yet to be discovered.

Intriguingly, the MGF110-7L gene was shown to be completely deleted within the genomes of both the naturally attenuated strain OURT88/3 and the cell-adapted strains ASFV-P121 and BA71V compared to the naturally virulent and/or parental strains, whereas this deletion was not observed in another cell-adapted strain, ASFV-G/VP110 (50–52). This feature suggests that MGF110-7L functions as a critical gene and is closely related to genetic variability and phenotypic alterations of ASFV. Similarly, deletion of the MGF110-9L gene was also observed during the adaptation of ASFV to grow in cultured cell lines. Of note, MGF110-9L was confirmed to play important roles in virus replication and virulence, and it has been targeted for live-attenuated vaccine development (33, 34). In contrast, deletion of the MGF110-1L, a conserved gene among different ASFV isolates, did not affect viral proliferation and virulence (53). In addition, the deletion of genes from MGF360 and MGF505/530 were also observed in the genomes of several attenuated ASFV isolates, including naturally occurring attenuated isolate and cell culture-adapted viruses (50–52). A number of reports have indicated that multiple members of MGF360 and MGF505/530 were implicated in the inhibition of type I interferon response and in virulence in pigs, and deletion of these genes could reduce replication in macrophages and attenuate virulence in pigs, an approach widely used for live-attenuated vaccine development and design (9–14). In combination with the correlation between genetic diversity and phenotypic changes during the process of ASFV adaptation, it is tempting to speculate that MGF110-7L plays significant roles in determining host range, viral replication, and virulence. Future studies are needed to confirm the functional relationship between the MGF110-7L gene and the biological properties of ASFV *in vitro* and *in vivo* in order to provide additional insight into how ASFV subverts host protein synthesis machinery under physiological conditions; such studies will contribute to the rational development of ASFV vaccines.

## MATERIALS AND METHODS

**Cells.** HEK293T (American Type Culture Collection [ATCC] CRL-11268), 3D4/21 (ATCC CRL-2843), and PK-15 (ATCC CCL-33) cells were grown at 37°C in a 5% $CO_2$ atmosphere culture in complete Dulbecco modified Eagle medium (DMEM; Gibco, Gaithersburg, MD) supplemented with 10% fetal bovine serum (FBS; Gibco), 100 U/mL penicillin, and 100 mg/mL streptomycin (Gibco).

**Antibodies and chemicals.** Rabbit anti-FLAG (80010-1-RR), anti-eIF2$\alpha$ (11170-1-AP), anti-p-eIF2$\alpha$ (28740-1-AP), anti-CHOP (15204-1-AP), and anti-ATF6 (24169-1-AP) polyclonal antibodies were purchased from Proteintech (Rosemont, IL). Rabbit anti-ATF4 (A0201), anti-G3BP1 (A3968), anti-BIP (A11366), anti-PERK (A18196), anti-PKR (A5578), anti-IRE1$\alpha$ (A17940), and anti-GCN2 (A7155) polyclonal antibodies; rabbit anti-p-PKR (AP1134) and anti-p-IRE1$\alpha$ (AP1146) monoclonal antibodies; and mouse anti-FLAG (AE005) and anti-$\beta$-actin

(AC026) monoclonal antibodies were purchased from ABclonal (Wuhan, China). Mouse anti-puromycin (MABE342) monoclonal antibody was purchased from Millipore (Billerica, MA). Rabbit anti-p-PERK (3179S) monoclonal antibody was purchased from Cell Signaling Technology (Danvers, MA). Rabbit anti-p-GCN2 (ab68427) monoclonal antibody was purchased from Abcam (Cambridge, MA). The secondary antibodies, conjugated with horseradish peroxidase (HRP), Alexa Fluor 488 (AF488), or Alexa Fluor 594 (AF594) were purchased from Invitrogen (Carlsbad, CA).

TG (HY-13433), ISRIB (HY-12495), and GSK2606414 (HY-18072) were purchased from MedChemExpress (Monmouth Junction, NJ). Puromycin (P8833), sodium arsenite (S7400), and PKR inhibitor C16 (I9785) were obtained from Sigma-Aldrich (St. Louis, MO). TG, ISRIB, GSK2606414, and PKR inhibitor C16 were dissolved in dimethyl sulfoxide, whereas puromycin and sodium arsenite was dissolved in water, to the required concentrations.

**Plasmid constructions and expression vectors.** All the ORFs of the ASFV strain China/2018/AnhuiXCGQ (GenBank accession number MK128995.1), including 11 members belonging to the MGF110, were generated by gene synthesis (GENEWIZ, Suzhou, China) and cloned into the pCMV vector with a C-terminal FLAG tag (Beyotime Biotechnology, Shanghai, China). The ATF4-RLuc and ATF4-EGFP reporters were constructed as previously described (31). Briefly, a full-length human ATF4 mRNA 5′ leader and ATF4 initiation codon were subcloned into pRL-TK (Promega, WI) and pEGFP-C1 (Clontech, Mountain View, CA), respectively. cDNAs of PDIA3, TMED4, and PSMA4 were amplified from total mRNA of PK-15 cells by conventional RT-PCR and inserted into the KpnI and XhoI sites of the pCAGGS-N-HA vector (Addgene, MA) to obtain hemagglutinin (HA)-tagged constructs. All constructed plasmids were verified by sequencing.

The organelle markers, including pDsRed2-ER, pDsRed2-Golgi, pDsRed2-Mito, pDsRed2-LAMP1, and pDsRed2-Peroxi vectors that encode targeting sequences of the corresponding organelles, were purchased from Clontech.

**Transient transfection.** A total of $3 \times 10^6$ 3D4/21 and PK-15 cells were seeded onto 12-well plates and incubated for 20 h until 70% confluent. Prior to transfection, the medium was changed to DMEM complete medium with 2% FBS. 3D4/21 and PK-15 cells were then transfected with a MGF110-7L-Flag-expressing plasmid with an increasing dose (0.5, 1.0, and 2.0 μg) or an empty Flag vector (2.0 μg) using Lipofectamine 2000 (Invitrogen, CA) according to the manufacturer's instructions. At 24 h posttransfection, the cells were treated as described and analyzed by Western blotting or RT-qPCR.

**Luciferase reporter assays.** HEK293T cells in 48-well plates were transfected with an expression plasmid for ASFV ORF (0.1 μg) and ATF4-RLuc plasmid (0.05 μg). At 24 h posttransfection, the cell extracts were harvested in a passive buffer, and a luciferase reporter assay system (Promega) was used to determine the RLuc activities according to the manufacturer's instructions.

**Cell viability assay.** After the cells were grown to 70 to 80% confluence in 12-well plates, they were transfected with pCMV-C-FLAG (vector, 2 μg) and MGF110-7L-FLAG plasmid (2 μg) for 24, respectively. Cells seeded into 96-well plates were incubated with each chemical at the indicated concentrations for 24 h. Cell viability was then measured using a CCK-8 assay (Beyotime Biotechnology, Shanghai, China) according to the manufacturer's instructions.

**Western blot analysis.** Unless indicated otherwise, protein samples for Western blot were prepared by direct lysis of the cells in 2× SDS sample buffer and boiled at 100℃ for 5 min. Samples were resolved by SDS-PAGE in an 8 or 12% acrylamide-bisacrylamide gel and transferred to a polyvinylidene difluoride membrane (Millipore, Billerica, MA). The membrane was blocked with 5% fat-free milk for 1 h at room temperature and then incubated with primary antibodies at 4℃ overnight. The following day, the membrane was washed three times with PBS-Tween and incubated with HRP-conjugated secondary antibody at a dilution of 1:4,000 for 1 h at room temperature. Finally, bands obtained after development with ECL reagent (Thermo Fisher Scientific, Waltham, MA) were visualized on a BioSpectrum 500 imaging system (UVP LLC, Upland, CA). The bands were quantified by densitometry and the data normalized to control values using ImageJ software.

**RT-qPCR.** Total RNA was extracted from 3D4/21 cells using the TRIzol reagent (Invitrogen) according to the manufacturer's instructions, and the cDNA synthesis was performed using the PrimeScript reverse transcriptase reagent kit (TaKaRa, Dalian, China). Quantitative PCR analysis was performed using TB Green Premix Ex Taq (TaKaRa) on an ABI QuantStudio 5 real-time PCR system (Agilent Technologies, Santa Clara, CA). The relative mRNA level of each gene was normalized to GAPDH mRNA levels and determined based on the standard $2^{-\Delta\Delta CT}$ protocol. The primers used for RT-qPCR were listed in Table 1.

**IFA and confocal microscopy.** PK-15 cells grown on glass coverslips were transfected with an empty Flag vector or a MGF110-7L-Flag-expressing vector until 40 to 50% confluence, or they were cotransfected with a FLAG-tagged plasmid and an indicated plasmid. At 24 h posttransfection, the cells were rinsed twice with phosphate-buffered saline (PBS), fixed with 4% paraformaldehyde for 15 min, washed with PBS, permeabilized with 0.1% Triton X-100 for 15 min, washed again with PBS, and blocked in 5% bovine serum albumin (Promega) in PBS for 1 h at room temperature. The cells were then incubated in primary antibodies (1:100) diluted in blocking buffer for 1 h at room temperature. After three washes with PBS-Tween, the cells were incubated with Alexa Fluor-conjugated secondary antibodies (1:500) diluted in blocking buffer for 1 h at room temperature. The cells were washed three times with PBS-Tween and incubated for 8 min with 4′,6′-diamidino-2-phenylindole (DAPI; Beyotime). The coverslips were then rinsed twice with PBS and Milli-Q water and mounted on cover slides using Antifade Mounting Medium (Beyotime). Then, confocal fluorescence images were visualized using a Zeiss LSM800 laser-scanning microscope (Carl Zeiss, Thornwood, NY).

**Ribopuromycylation assay.** The ribopuromycylation assay was performed as previously describe (38). Briefly, the cells were transfected with the MGF110-7L-Flag-expressing vector at the indicated dose or with an empty Flag vector. At 24 h posttransfection, the cells were pulse-labeled with 5 μg/mL puromycin and then incubated for 30 min. As a positive control, cells transfected with an empty vector were

**TABLE 1** RT-qPCR primers used in this study

| Target | Orientation | Sequence (5′–3′) |
|---|---|---|
| Porcine ATF4 | Forward | CCCTTTACGTTCTTGCAAACTC |
| | Reverse | GCTTCCTATCTCCTTCCGAGA |
| Porcine CHOP | Forward | CTCAGGAGGAAGAGGAGGAAG |
| | Reverse | GCTAGCTGTGCCACTTTCCTT |
| Porcine GADD34 | Forward | AAGAGCCTGGAGAGAGGGAGAG |
| | Reverse | GTCCCCAGGTTTCCAAAAGCA |
| Porcine XBP1(s) | Forward | GAGTCCGCAGCAGGTG |
| | Reverse | CCGTCAGAATCCATGGGG |
| Porcine XBP1(u) | Forward | TCCGCAGCACTCAGACTACGT |
| | Reverse | ATGCCCAAGAGGATATCAGACTC |
| Porcine ERdj4 | Forward | CAGAGAGATTGCAGAAGCATATGA |
| | Reverse | GCTTCTTGGATCGAGTGTTTT |
| Porcine P58IPK | Forward | GGTGCTGAATGTGGAGTAAAT |
| | Reverse | GCATGAAACTGAGATAAAGCG |
| Porcine EDEM1 | Forward | GAGGCATGTTCGTCTTCGG |
| | Reverse | CGGCAGTGGATGGGGTTGAG |
| Porcine BIP | Forward | CATCACGCCGTCATATGTGG |
| | Reverse | CGTCGAAGACCGTGTTCTCA |
| Porcine GRP94 | Forward | GCTGAGGATGAAGTGGATGTGG |
| | Reverse | CATCTGTCCTGGAACCTTCTCTA |
| Porcine calreticulin | Forward | CCCACTATTTACTTCAAGGAG |
| | Reverse | GAATTTGCCGGAACTGAGAAC |
| Porcine GAPDH | Forward | ACATGGCCTCCAAGGAGTAAGA |
| | Reverse | GATCGAGTTGGGGCTGTGACT |

treated with 0.5 mM arsenite for 45 min before the addition of puromycin. After three washes with PBS, the cells were prepared and subjected to IFA or Western blot analysis.

**Structure and function prediction of MGF110-7L protein.** Protein sequence within ASFV strain China/2018/AnhuiXCGQ (MH766894.1) was selected as a query to predict the structure and function features of MGF110-7L protein using various software tools which were listed in Table 2.

**Co-IP.** HEK293T cells with plasmids transfection were lysed in NP-40 lysis buffer supplemented with protease inhibitor cocktail (Roche, Basel, Switzerland) and sonicated for 1.5 min with an interval of 5 s. After centrifugation (15,000 × *g* for 20 min), supernatants were incubated with the indicated antibody or control IgG at 4℃ overnight. Immunoprecipitated complexes were then mixed with protein G agarose beads (Roche) at 4℃ for 3 h. After three washes with ice-cold lysis buffer, beads were boiled in 1× SDS sample buffer and analyzed by Western blotting.

**IP and proteomic analysis.** HEK293T cells were transfected with an empty Flag vector or a MGF110-7L-Flag-expressing vector. After 24 h, cells were lysed with NP-40 lysis buffer, and cell lysate was subjected to Flag immunoprecipitation as described above. The proteins bound to the beads were visualized by silver staining or Western blot. Then, the protein bands corresponding to the empty vector and MGF110-7L-Flag pulldown lanes that differed in staining intensity were excised from the gel and subsequently subjected to in-gel tryptic digestion to extract peptides for further LC-MS/MS analysis. LC-MS/MS analysis was performed as described previously (54). After the monoisotopic mass lists for each protonated peptide were obtained, protein identification and

**TABLE 2** Software tools for predicting MGF110-7L protein structure and functions

| Service | Website | Function (reference) |
|---|---|---|
| ClustalX 2.1 | http://www.ebi.ac.uk/tools/clustalw2 | Multiple sequence alignment program (55) |
| ESPript 3.0 | http://espript.ibcp.fr/ESPript/cgi-bin/ESPript.cgi | Generate a pretty PostScript output from aligned sequences (56) |
| NetSurfP 2.0 | https://services.healthtech.dtu.dk/service.php?NetSurfP-2.0 | Predict secondary structure for each residue of the input sequences (57) |
| ProtScale | https://web.expasy.org/protscale/ | Predict hydrophobicity or hydrophilicity scales on a selected protein |
| SignalP 5.0 | https://services.healthtech.dtu.dk/service.php?SignalP-5.0 | Predict the presence of signal peptides and the location of their cleavage sites in proteins (58) |
| NetNGlyc | https://services.healthtech.dtu.dk/service.php?NetNGlyc-1.0 | Predict N-glycosylation sites in proteins (59) |
| TMHMM-2.0 | https://services.healthtech.dtu.dk/service.php?TMHMM-2.0 | Predict the topological structure of transmembrane proteins (60) |
| I-TASSER | https://zhanggroup.org/I-TASSER/ | Protein structure prediction and structure-based function annotation (61) |

quantification were carried out using Mascot 2.3.02 (MatrixScience, London, UK) against the Swiss-Prot *Homo sapiens* sequence database. All experiments were carried out with two independent biological replicates.

**Statistical analysis.** The experimental data are presented in graph bars as means ± the standard deviations (SD) of at least three independent biological replicates. Statistical significance was determined using a Student $t$ test by SPSS Statistics (IBM Corporation, USA); significance is indicated in the figures by asterisks (*, $P < 0.05$; **, $P < 0.01$; ns, not significant).

## SUPPLEMENTAL MATERIAL

Supplemental material is available online only.

**SUPPLEMENTAL FILE 1**, XLSX file, 0.02 MB.

## ACKNOWLEDGMENTS

This project was supported by the National Natural Science Foundation of China (grants 31941001 and 32002278), the China Postdoctoral Science Foundation (grant 2021M701104), and the Key R&D Program of Henan Province (grant 212102110369).

Conceptualization—S.H., B.W., and G.Z.; methodology—S.H., H.Z., and S.F.; project administration—H.Z., S.F., Y.Z., Y.D., A.Z., and D.J.; writing (original draft)—S.H., H.Z., and S.F.; writing (review and editing)—S.H., B.W., and G.Z. All authors heave read and approved the final submitted manuscript.

We declare there are no conflicts of interest.

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
