## [Reviewer comments · Microbiology Spectrum]

Microbiology Spectrum

African swine fever virus MGF110-7L induces host cell translation suppression and stress granule formation by activating the PERK/PKR-eIF2 α pathway

Han Zhong, Shuai Fan, Yongkun Du, Yuhang Zhang, Angke Zhang, Dawei Jiang, Shichong Han, Bo Wan, and Gaiping Zhang

Corresponding Author(s): Shichong Han, College of Veterinary Medicine, Henan Agricultural University

Review Timeline:

Submission Date:	August 18, 2022
Editorial Decision:	September 26, 2022
Revision Received:	October 20, 2022
Accepted:	October 30, 2022

Editor: Manjula Kalia

Reviewer(s): Disclosure of reviewer identity is with reference to reviewer comments included in decision letter(s). The following individuals involved in review of your submission have agreed to reveal their identity: Fernando Ferreira (Reviewer #3)

Transaction Report:

DOI: <https://doi.org/10.1128/spectrum.03282-22>

September 26, 2022

Dr. Shichong Han
College of Veterinary Medicine, Henan Agricultural University
Zhengzhou, Henan 450046
China

Re: Spectrum03282-22 (African swine fever virus MGF110-7L induces host cell translation suppression and stress granule formation by activating the PERK/PKR-eIF2 α pathway)

Dear Dr. Shichong Han:

Thank you for submitting your manuscript to Microbiology Spectrum. Your manuscript has been reviewed by two experts and both have appreciated the study. The reviewer's have made minor suggestions. I invite you to address their comments and submit a revised manuscript.

Link Not Available

Sincerely,

Manjula Kalia

Journals Department
Reviewer comments:

Reviewer #1 (Comments for the Author):

Comments on Zhong et al.,

The study by Zhong et al has dissected how ASFV infection controls and subverts the host translational machinery. Here, the authors have identified and characterized the role of MGF110-7L, an uncharacterized member of the multigene family 110, in host translation regulation. MGF110-7L is seen to express throughout the virus life cycle and is assumed to play a crucial role in ASFV pathogenesis. ASFV MGF110-7L is seen to induce eif2 α phosphorylation which leads to the suppression of host translation and enhancement of stress granule formation by activating upstream PKR and PERK dependent pathway. The study is well designed and the manuscript is also very well written.

However, I have a few queries as follows

1. In Fig. 1A, please provide the statistical data on RLuc activity in presence of all the tested plasmids, as shown in the presence of MGF110-7L.
2. Fig. 1A legend, please mention the concentration of different vectors expressing different members of the MGF110 family used in the experiment.
3. It is unclear how the % ATF4-EGFP expression is measured or calculated in Fig. 1B. Is this % of cells expressing EGFP or the total % of EGFP expression? Please elaborate on how the analysis is done and the percentage is calculated in the figure legend or material and methods section.
4. In Fig.1D, WB shows relatively higher expression of ATF4 in vector alone than 0.5/1 ug MGF110-7L transfected cells whereas the beta-actin seems similar, please explain or provide a low exposure image of beta-actin.
5. In Fig. 3 B and C, the effect of eif2 α phosphorylation in SG formation upon arsenite treatment was reversed using DP71L whereas MGF110-7L induced SG formation was reversed using ISRIB. Ideally, the same inhibitor should be used in positive and test treatment. Please explain.
6. Please provide the intensity band ratios for Fig. E and F.
7. In line 274, the authors have mentioned that MGF-110 7L barely colocalizes with peroxi- marker, which is confusing to interpret. The fluorescence intensity graph seems to suggest partial localization, similar to Golgi marker. Please explain and also clear in the text.
8. In line 264, the heading suggests the involvement of ASFV MGF 110-7L in the secretory pathway, whereas no direct experiments have been done to prove the same. Localization of MGF 110-7L with different secretory compartments doesn't clearly suggest its involvement/role in the pathway. The heading (Line 264) should be modified. However, the possible reasons and functions related to their cellular distribution can be extrapolated in the results/discussion.
9. Authors have not mentioned anywhere the no. of biological replicates or how many times the Co-IP experiments have been repeated to conclude the MGF 110-7L host interactome shown in Fig. 7. It seems the data is obtained from only one biological experiment/ sample. Please provide detailed information about how the final list of interacting proteins is obtained for GO analysis.
10. Line 287, suggests the involvement of ASFV MGF110-7L in the modulation of ER redox homeostasis, etc., whereas authors have identified the MGF110-7L host interactors which are involved in ER redox homeostasis. The establishment of the direct involvement of MGF110-7L in ER-redox homeostasis needs further detailed experiments. Please modify the heading.
11. Please provide the species-specific information in table 1.
12. Please provide the concentration of PDIA3-HA, TMED4-HA, or PSMA4-HA plasmid in Fig. 8 legend. Please also make sure that the concentration of all the plasmids used in the study is provided in the manuscript.

Reviewer #3 (Comments for the Author):

The manuscript submitted by Shichong Han et al. entitled "African swine fever virus MGF110-7L induces host cell translation suppression and stress granule formation by activating the PERK/PKR-eIF2 α pathway" aims to evaluate and characterize the role of an ASFV viral protein in the infection cycle.

In the Introduction section the authors wrote that "there are still no protective vaccines" which is not true. They should correct this point and add a very recent reference about ASFV vaccines (10.1080/22221751.2022.2108342). Additionally, the authors should briefly introduce some knowledge already known about the ASFV RNA helicases (e.g. 10.1080/22221751.2019.1578624).

In the results's section, authors should present the major limitations of using HEK293T cells to quantify the regulation of the regulation of translation rate of ATF4-RLuc, since the ASFV replication primarily occurs in macrophages.

In fig 1. please identify the different graphs, some are missing.

Staff Comments:

Preparing Revision Guidelines

Please return the manuscript within 60 days; if you cannot complete the modification within this time period, please contact me. If you do not wish to modify the manuscript and prefer to submit it to another journal, please notify me of your decision immediately so that the manuscript may be formally withdrawn from consideration by Microbiology Spectrum.

Comments on Zhong et al.,

The study by Zhong et al has dissected how ASFV infection controls and subverts the host translational machinery. Here, authors have identified and characterized the role of MGF110-7L, an uncharacterized member of the multigene family 110, in host translation regulation. MGF110-7L is seen to express throughout the virus life cycle and is assumed to play a crucial role in ASFV pathogenesis. ASFV MGF110-7L is seen to induce eif2 α phosphorylation which leads to the suppression of host translation and enhancement of stress granule formation by activating upstream PKR and PERK dependent pathway. The study is well designed and the manuscript is also very well written.

However, I have a few queries as follows

1. In Fig. 1A, please provide the statistical data on RLuc activity in presence of all the tested plasmids, as shown in the presence of MGF110-7L.
2. Fig. 1A legend, please mention the concentration of different vectors expressing different members of the MGF110 family used in the experiment.
3. It is unclear how the % ATF4-EGFP expression is measured or calculated in Fig. 1B. Is this % of cells expressing EGFP or the total % of EGFP expression? Please elaborate on how the analysis is done and the percentage is calculated in the figure legend or material and methods section.
4. In Fig.1D, WB shows relatively higher expression of ATF4 in vector alone than 0.5/1 ug MGF110-7L transfected cells whereas the beta-actin seems similar, please explain or provide a low exposure image of beta-actin.
5. In Fig. 3 B and C, the effect of eif2 α phosphorylation in SG formation upon arsenite treatment was reversed using DP71L whereas MGF110-7L induced SG formation was reversed using ISRIB. Ideally, the same inhibitor should be used in positive and test treatment. Please explain.
6. Please provide the intensity band ratios for Fig. E and F.
7. In line 274, the authors have mentioned that MGF-110 7L barely colocalizes with peroxi- marker, which is confusing to interpret. The fluorescence intensity graph seems to suggest partial localization, similar to Golgi marker. Please explain and also clear in the text.

8. In line 264, the heading suggests the involvement of ASFV MGF 110-7L in the secretory pathway, whereas no direct experiments have been done to prove the same. Localization of MGF 110-7L with different secretory compartments doesn't clearly suggest its involvement/role in the pathway. The heading (Line 264) should be modified. However, the possible reasons and functions related to their cellular distribution can be extrapolated in the results/discussion.
9. Authors have not mentioned anywhere the no. of biological replicates or how many times the Co-IP experiments have been repeated to conclude the MGF 110-7L host interactome shown in Fig. 7. It seems the data is obtained from only one biological experiment/ sample. Please provide detailed information about how the final list of interacting proteins is obtained for GO analysis.
10. Line 287, suggests the involvement of ASFV MGF110-7L in the modulation of ER redox homeostasis, etc., whereas authors have identified the MGF110-7L host interactors which are involved in ER redox homeostasis. The establishment of the direct involvement of MGF110-7L in ER-redox homeostasis needs further detailed experiments. Please modify the heading.
11. Please provide the species-specific information in table 1.
12. Please provide the concentration of PDIA3-HA, TMED4-HA, or PSMA4-HA plasmid in Fig. 8 legend. Please also make sure that the concentration of all the plasmids used in the study is provided in the manuscript.

Responses to Reviewers' Comments on Manuscript ID Spectrum03282-22

We sincerely thank anonymous reviewers for their vigorous and thoughtful comments of the manuscript. We have worked seriously with the comments and believe that this version of the manuscript is greatly improved.

Reviewer comments:

Reviewer #1 (Comments for the Author):

The study by Zhong et al has dissected how ASFV infection controls and subverts the host translational machinery. Here, the authors have identified and characterized the role of MGF110-7L, an uncharacterized member of the multigene family 110, in host translation regulation. MGF110-7L is seen to express throughout the virus life cycle and is assumed to play a crucial role in ASFV pathogenesis. ASFV MGF110-7L is seen to induce eIF2 α phosphorylation which leads to the suppression of host translation and enhancement of stress granule formation by activating upstream PKR and PERK dependent pathway. The study is well designed and the manuscript is also very well written.

Specific Comments:

1. In Fig. 1A, please provide the statistical data on RLuc activity in presence of all the tested plasmids, as shown in the presence of MGF110-7L.

*Response: We greatly appreciate your valuable comment. Based on your suggestion, we showed the statistical data on RLuc activity in presence of all the tested plasmids in the revised Fig. 1A (see below). The data showed as mean \pm standard deviation (SD) of three independent experiments. Significance was determined with Student's t test by SPSS Statistics (IBM Corporation, USA) and was assigned at *P < 0.05 and **P < 0.01.*

*In addition, we also presented the ATF4-RLuc activity upon ASFV DP71L transfection in the revised Fig. 1A (see below), and modified the text in the revised manuscript. **Please see page 7 lines 147-155 in the Results section:** "In TG-treated group, expression of the ATF4-RLuc was increased ~8 times greater than basal level; whereas the ASFV DP71L, which has been previously shown to prevent the phosphorylation of eIF2 α and the induction of ATF4 (24), was found to remarkably inhibit ATF4-RLuc expression. These findings supported the validity of our experimental approach (Fig. 1A). In these screens, most members of ASFV MGF110 family were identified to induce the translational expression of ATF4-RLuc, and MGF110-7L and MGF110-9L showed the strong stimulatory effects with higher than 5-fold up-regulation."*

2. Fig. 1A legend, please mention the concentration of different vectors expressing different members of the MGF110 family used in the experiment.

Response: We greatly appreciate the critical comment. Based on your suggestion, we added the concentration of different vectors expressing different members of the MGF110 family in the revised Fig. 1A legend (see page 36 lines 838-844). In addition, we carefully checked the whole manuscript, and we ensured that the concentrations of all the plasmids used in the study were shown in the revised Figure Legends section (see pages 36-41 lines 837-983).

3. It is unclear how the % ATF4-EGFP expression is measured or calculated in Fig. 1B. Is this % of cells expressing EGFP or the total % of EGFP expression? Please elaborate on how the analysis is done and the percentage is calculated in the figure legend or material and methods section.

Response: We greatly appreciate your constructive comments. Based on your suggestions, we have elaborated on how the quantification of ATF4-EGFP expression is done and the percentage is calculated in the revised Fig. 1B legend. Please see page 36 lines 847-850 as follows: "Quantification of ATF4-EGFP expression was done using ImageJ software in at least 10 random fields of view with greater than 800 cells analysed on each slide. Bar graphs on the right show the percentage of cells expressing EGFP in each group under different treatments."

4. In Fig.1D, WB shows relatively higher expression of ATF4 in vector alone than 0.5/1 ug MGF110-7L transfected cells whereas the beta-actin seems similar, please explain or provide a low exposure image of beta-actin.

Response: Thank you very much for pointing out the ambiguous immunoblots regarding ATF4 and β -actin. Overexposure of immunoblots may affect the research conclusions. Based on your suggestions, we have carefully checked all the previous bands and repeated the relevant experiments. The new immunoblots also confirmed that ectopic expression of MGF110-7L triggered a significant increase in the levels of P-eIF2 α and ATF4 protein in a dose-dependent manner, which is consistent with the research conclusions.

In addition, we replaced the previous Fig. 1D with a version of better quality (see the new Fig. 1D below).

5. In Fig. 3 B and C, the effect of eIF2α phosphorylation in SG formation upon arsenite treatment was reversed using DP71L whereas MGF110-7L induced SG formation was reversed using ISRIB. Ideally, the same inhibitor should be used in positive and test treatment. Please explain.

Response: We greatly appreciate your constructive comments. By taking heart your suggestions, we have performed additional experiments (arsenite + ISRIB) and modified the figure and text in the revised manuscript (see page 10 lines 224-235). As shown in revised Fig. 3B and C (see below), ISRIB was found to result in a dramatic reduction of arsenite-induced SGs, with ~13% of cells containing SGs. Moreover, the ASFV DP71L, an eIF2α phosphorylation antagonists, also blocked arsenite-induced SGs assembly. These findings indicated that the phosphorylation status of eIF2α is positively correlated to SGs formation. Importantly, ISRIB treatment of MGF110-7L-expressing cells also substantially decreased the amount of SGs compared to untreated cells, with only 2.8% of cells containing SGs. Altogether, these results reveal that MGF110-7L induces SGs formation and the process is dependent upon the phosphorylation of eIF2α.

6. Please provide the intensity band ratios for Fig. 4E and F.

*Response: We appreciate the valuable comment. Based on your suggestions, we have determined the grayscale values of the protein bands in Fig. 4E and F using Image J software. Relative amount of p-PERK, p-PKR, or p-eIF2 α in each sample after normalizing to the corresponding total PERK, total PKR, or total p-eIF2 α was calculated and plotted in bar graphs for comparison the intensity band ratios. Data are presented as the means \pm SD of the results of three independent experiments (**, $P < 0.01$; n.s., not significant.).*

In addition, we have modified the Fig. 4E and F and the corresponding legend in the revised manuscript (see below).

Figure legend. (E and F) 3D4/21 cells were transfected with an empty Flag vector or a MGF110-7L-Flag expressing vector (2.0 μ g), with or without GSK2606414 (10 μ M) or C16 (1 μ M) as indicated for 24 h before cell lysate samples were obtained. Lysates were analyzed via immunoblotting with the indicated antibodies. The relative levels of p-PERK, p-PKR, or p-eIF2 α in each sample after normalizing to the corresponding total PERK, total PKR, or total p-eIF2 α was determined using Image J software and plotted in bar graphs. Data in panels B-D are the means \pm SD of the results of three independent experiments. **, $P < 0.01$.

7. In line 274, the authors have mentioned that MGF-110 7L barely colocalizes with peroxi-marker, which is confusing to interpret. The fluorescence intensity graph seems to suggest partial localization, similar to Golgi marker. Please explain and also clear in the text.

Response: We greatly appreciate your constructive comments. Based on your suggestions, we have carefully determined the overlap coefficient (R) and colocalization profile between MGF110-7L and a peroxisomal marker using ImageJ software in more than 3 random fields of view. As shown in Figure below, the colocalization analysis results indicated that MGF-110 7L partially co-localized with the peroxisomal marker, which is consistent with your opinions.

Figure legend. PK-15 cells in 12-well plates were cotransfected with a MGF110-7L-Flag expressing vector (0.5 μ g) and the peroxisome marker pDsRed2-Peroxi (0.2 μ g). At 24 h posttransfection, the cells were fixed, permeabilized and incubated with anti-FLAG antibody and then with secondary antibody conjugated with AF488 (green). Nuclei were counterstained with DAPI (blue). The peroxisome marker was directly visualized (red), and localization was determined using confocal microscopy. The overlapping coefficient (R) was shown in enlarged images, and the intensity profile of the linear region of interest (ROI) across the PK-15 cell contained with MGF110-7L and peroxisome marker. Scale bar, 20 μ m.

Thanks again for your careful review and for pointing out these mistakes. Also, we have modified the Fig. 6 and the relevant sentences in the revised manuscript. Please see page 12 lines 280-283 in Results section: "As shown in Fig. 6, MGF110-7L was diffusely distributed in the cells and colocalized predominantly with the ER marker, whereas a partial colocalization was observed between MGF110-7L and the marker of Golgi apparatus, mitochondria, lysosome, or peroxisome."; and please see page 18 lines 427-430 in Discussion section: "Using confocal immunofluorescence technique, MGF110-7L was confirmed to be mainly located in the ER lumen and that a small amount is retained in other intracellular organelles, including Golgi apparatus, mitochondria, lysosome, and peroxisome."

8. In line 264, the heading suggests the involvement of ASFV MGF 110-7L in the secretory pathway, whereas no direct experiments have been done to prove the same. Localization of MGF 110-7L with different secretory compartments doesn't clearly suggest its involvement/role in the pathway. The heading (Line 264) should be modified. However, the possible reasons and functions related to their cellular distribution can be extrapolated in the results/discussion. *Response: We greatly appreciate your constructive comments. We totally agree that localization of MGF 110-7L with different secretory compartments doesn't clearly suggest its involvement/role in the secretory pathway. Therefore, we have changed the heading (see page 12 line 274) in the revised version: "ASFV MGF110-7L is primarily located in the ER."*

Meanwhile, the possible reasons and functions related to their cellular distribution were extrapolated in the Results/Discussion section. **Please see page 12 lines 291-295 in Results section:** “Taken together, these results indicated that MGF110-7L is predominantly located in the ER and a small amount is retained in other intracellular compartments within the secretory pathway, along with disruption of the structural components of specific organelles, suggesting the involvement of MGF 110-7L in the secretory pathway.”; **and please see page 18 lines 430-434 in Discussion section:** “MGF110-7L also triggers a significant reorganization of the subcellular distribution and morphological characteristics of the Golgi and peroxisome, suggesting that this protein plays important roles in the process of remodeling ER/Golgi apparatus, protein sorting at the ER–Golgi interface, and peroxisome generation.”.

9. Authors have not mentioned anywhere the no. of biological replicates or how many times the Co-IP experiments have been repeated to conclude the MGF 110-7L host interactome shown in Fig. 7. It seems the data is obtained from only one biological experiment/ sample. Please provide detailed information about how the final list of interacting proteins is obtained for GO analysis.

Response: We greatly appreciate your valuable comments, and we would like to making an explanation regarding this issue. We conducted the Co-IP experiments with two independent biological replicates (see Figure blow). Then, the protein bands corresponding to the empty vector and MGF110-7L-Flag pulldown lanes that differed in staining intensity were excised from the gel and subsequently subjected to LC–MS/MS analysis, and protein identification was carried out using Mascot 2.3.02 (MatrixScience, London, UK) against the Swiss-Prot Homo sapiens sequence database.

*In addition, based on your suggestions, we have added this information in the revised manuscript. **Please see page 13 lines 303-306 in Results section:** “We carried out the experiments with two independent biological replicates. The overview of the identified 81 cellular proteins interacting with MGF110-7L from two quantitative mass spectrometry analysis was presented in Table S1.” **Please see page 25 lines 612-613 in Materials and Methods***

section: “All the experiments were carried out with two independent biological replicates.”
Please see page 40 lines 939-940 in Figure Legends section: “The data were tested two times independently.”

10. Line 287, suggests the involvement of ASFV MGF110-7L in the modulation of ER redox homeostasis, etc., whereas authors have identified the MGF110-7L host interactors which are involved in ER redox homeostasis. The establishment of the direct involvement of MGF110-7L in ER-redox homeostasis needs further detailed experiments. Please modify the heading.

Response: We greatly appreciate your constructive comments. We totally agree that the establishment of the direct involvement of MGF110-7L in ER-redox homeostasis needs further detailed experiments. Based on your suggestion, we have changed the heading (see page 13 lines 296-297) in the revised version: “Identification of ASFV MGF110-7L-interacting host factors involved in modulation of ER redox homeostasis.”

11. Please provide the species-specific information in table 1.

Response: We appreciate the valuable comment. Based on your suggestion, we added the species-specific information of these RT-qPCR primers in revised Table 1 (see below).

TABLE 1 RT-qPCR primers used in this study

Target	Orientation	Sequence
Porcine ATF4	Forward	5'-CCCTTTACGTTCTTGCAAACCTC-3'
	Reverse	5'-GCTTCCTATCTCCTTCCGAGA-3'
Porcine CHOP	Forward	5'-CTCAGGAGGAAGAGGAGGAAG-3'
	Reverse	5'-GCTAGCTGTGCCACTTTTCCTT-3'
Porcine GADD34	Forward	5'-AAGAGCCTGGAGAGAGGAGAG-3'
	Reverse	5'-GTCCCCAGGTTTCCAAAAGCA-3'
Porcine XBP1(s)	Forward	5'-GAGTCCGCAGCAGGTG-3'
	Reverse	5'-CCGTCAGAATCCATGGGG-3'
Porcine XBP1(u)	Forward	5'-TCCGCAGCACTCAGACTACGT-3'
	Reverse	5'-ATGCCCAAGAGGATATCAGACTC-3'
Porcine ERdj4	Forward	5'-CAGAGAGATTGCAGAAGCATATGA-3'
	Reverse	5'-GCTTCTTGGATCGAGTGTTTT-3'
Porcine P58IPK	Forward	5'-GGTGTGAATGTGGAGTAAAT-3'
	Reverse	5'-GCATGAAACTGAGATAAAGCG-3'
Porcine EDEM1	Forward	5'-GAGGCATGTTTCGTCTTCGG-3'
	Reverse	5'-CGGCAGTGGATGGGGTTGAG-3'
Porcine BIP	Forward	5'-CATCACGCCGTCATATGTGG-3'
	Reverse	5'-CGTCGAAGACCGTGTTCTCA-3'
Porcine GRP94	Forward	5'-GCTGAGGATGAAGTGGATGTGG-3'
	Reverse	5'-CATCTGTCCTGGAACCTTCTTA-3'
Porcine Calreticulin	Forward	5'-CCCCTATTACTTCAAGGAG-3'
	Reverse	5'-GAATTTGCCGGAAGTGAAG-3'
Porcine GAPDH	Forward	5'-ACATGGCCTCCAAGGAGTAAGA-3'
	Reverse	5'-GATCGAGTTGGGGCTGTGACT-3'

12. Please provide the concentration of PDIA3-HA, TMED4-HA, or PSMA4-HA plasmid in Fig. 8 legend. Please also make sure that the concentration of all the plasmids used in the study is provided in the manuscript.

Response: We greatly appreciate the critical comments. Based on your suggestions, we added the concentration of PDIA3-HA, TMED4-HA, or PSMA4-HA plasmid in the revised Fig. 8 legend (see page 40 lines 951-962). In addition, we carefully checked the whole manuscript, and we ensured that the concentrations of all the plasmids used in the study were shown in the revised Figure Legends section (see page 36-41 lines 837-983).

Reviewer #3 (Comments for the Author):

The manuscript submitted by Shichong Han et al. entitled "African swine fever virus MGF110-7L induces host cell translation suppression and stress granule formation by activating the PERK/PKR-eIF2 α pathway" aims to evaluate and characterize the role of an ASFV viral protein in the infection cycle.

Specific Comments:

1. In the Introduction section the authors wrote that "there are still no protective vaccines" which is not true. They should correct this point and add a very recent reference about ASFV vaccines (10.1080/22221751.2022.2108342). Additionally, the authors should briefly introduce some knowledge already known about the ASFV RNA helicases (e.g. 10.1080/22221751.2019.1578624).

Response: We thank the reviewer for these valuable comments. Based on your suggestions, we have revised the text and added the relevant references in the Introduction section. **Please see page 4 lines 74-80 as follows:** "The spread of ASF has posed a devastating burden on pig industry and a huge socio-economic impact in the worldwide. Currently, implementing strict biosecurity measures and culling of infected herds are the main ways to control its spreading since there are no safe and effective vaccines or targeted therapeutics. However, the continuous deterioration of ASF shows these control strategies to be lacking, and the lack of vaccines or therapeutic options warrants urgent further investigation (1)."; **and please see page 4 lines 85-95 as follows:** "The virus genome is a linear double-stranded DNA molecule of ~170 to 194 kbp encoding more than 150 open reading frames (ORFs). Genomic variation between strains are largely due to gain or loss of genes from the multigene families (MGFs) (3). So far, several ASFV proteins have been reported to play important roles in multiple stages of viral infection including the transcription and translation, morphogenesis, immune escape, etc (4-6). For example, the ASFV QP509L and Q706L RNA helicases are mainly involved in viral transcription events (6, 7). The ASFV pH240R is required for the efficient production of infectious progeny virions (8). Notably, the members of MGF360 and MGF530/505 are confirmed to be IFN antagonists and play important roles in virus replication, virulence, pathogenicity, and host range (9-14).".

References

1. Urbano AC, Ferreira F. 2022. **African swine fever control and prevention: an update on vaccine development.** *Emerg Microbes Infect* 11:2021-2033.
3. Dixon LK, Chapman DA, Netherton CL, Upton C. 2013. **African swine fever virus replication and genomics.** *Virus Res* 173:3-14.
4. Alejo A, Matamoros T, Guerra M, Andres G. 2018. **A proteomic atlas of the African**

- swine fever virus particle.** *J Virol* 92:e01293-18.
5. Wang Y, Kang W, Yang W, Zhang J, Li D, Zheng H. 2021. **Structure of African swine fever virus and associated molecular mechanisms underlying infection and immunosuppression: a review.** *Front Immunol* 12:715582.
 6. Simoes M, Freitas FB, Leitao A, Martins C, Ferreira F. 2019. **African swine fever virus replication events and cell nucleus: new insights and perspectives.** *Virus Res* 270:197667.
 7. Freitas FB, Frouco G, Martins C, Ferreira F. 2019. **The QP509L and Q706L superfamily II RNA helicases of African swine fever virus are required for viral replication, having non-redundant activities.** *Emerg Microbes Infect* 8:291-302.
 8. Zhou P, Li LF, Zhang K, Wang B, Tang L, Li M, Wang T, Sun Y, Li S, Qiu HJ. 2021. **Deletion of the H240R gene of African swine fever virus decreases infectious progeny virus production due to aberrant virion morphogenesis and enhances the inflammatory cytokines expression in porcine macrophages.** *J Virol* 96:e0166721.
 9. Golding JP, Goatley L, Goodbourn S, Dixon LK, Taylor G, Netherton CL. 2016. **Sensitivity of African swine fever virus to type I interferon is linked to genes within multigene families 360 and 505.** *Virology* 493:154-161.
 10. O'Donnell V, Holinka LG, Gladue DP, Sanford B, Krug PW, Lu X, Arzt J, Reese B, Carrillo C, Risatti GR, Borca MV. 2015. **African swine fever virus Georgia isolate harboring deletions of MGF360 and MGF505 genes is attenuated in swine and confers protection against challenge with virulent parental virus.** *J Virol* 89:6048-6056.
 11. Rathakrishnan A, Connell S, Petrovan V, Moffat K, Goatley LC, Jabbar T, Sánchez-Cordón PJ, Reis AL, Dixon LK. 2022. **Differential effect of deleting members of African swine fever virus multigene families 360 and 505 from the genotype II Georgia 2007/1 isolate on virus replication, virulence, and induction of protection.** *J Virol* 96:e0189921.
 12. Li J, Song J, Kang L, Huang L, Zhou S, Hu L, Zheng J, Li C, Zhang X, He X, Zhao D, Bu Z, Weng C. 2021. **pMGF505-7R determines pathogenicity of African swine fever virus infection by inhibiting IL-1beta and type I IFN production.** *PLoS Pathog* 17:e1009733.
 13. Zhang K, Yang B, Shen C, Zhang T, Hao Y, Zhang D, Liu H, Shi X, Li G, Yang J, Li D, Zhu Z, Tian H, Yang F, Ru Y, Cao WJ, Guo J, He J, Zheng H, Liu X. 2022. **MGF360-9L is a major virulence factor associated with the African swine fever virus by antagonizing the JAK/STAT signaling pathway.** *mBio* 13:e0233021.
 14. Zsak L, Lu Z, Burrage TG, Neilan JG, Kutish GF, Moore DM, Rock DL. 2001. **African swine fever virus multigene family 360 and 530 genes are novel macrophage host range determinants.** *J Virol* 75:3066-3076.

2. In the results's section, authors should present the major limitations of using HEK293T cells to quantify the regulation of translation rate of ATF4-RLuc, since the ASFV replication primarily occurs in macrophages.

Response: We greatly appreciate your valuable comments, and we would like to make several clarifications regarding this issue.

First, HEK293T cell is a cell line exhibiting epithelial morphology that was isolated from human embryo kidney tissue. The cell line is a derivative of the 293T (293tsA1609neo) cell line, a highly transfectable derivative of the 293 cell line into which the temperature sensitive gene

for SV40 T-antigen was inserted. Given its high transfectability, HEK293T cell has been widely used for transient protein expression and high-throughput screening. Notably, HEK293T cell has also been successfully used for preliminary screening of ASFV genes based on the genetic reporter systems. Second, since the ASFV replication primarily occurs in macrophages, we further investigated the involvement of ASFV MGF110-7L in eIF2 α signaling in PK-15 cells and porcine alveolar macrophages-derived 3D4/21 cells, which supported the replication of cell-adapted or field isolates of ASFV (36). These biochemical results showed that the ectopic expression of MGF110-7L remarkably increased eIF2 α phosphorylation and ATF4 translation in a dose-dependent manner in both cell lines, which supported the validity of our screening results. Third, after continuous passaging, the adapted ASFV strain can replicate efficiently in HEK293T cells (52). For the above reasons, HEK293T cells were used to screen the ASFV proteins that regulate translation rate of ATF4-RLuc in this study.

In addition, based on your suggestions, we have revised the text in the Results section. Please see page 6 lines 140-146: “To identify ASFV proteins that induce eIF2 α phosphorylation, we constructed expression plasmids for 179 ORFs of the ASFV China/2018/AnhuiXCGQ strain and ATF4- Renilla luciferase (RLuc) by replacing the main coding region of ATF4 with RLuc. We next examined their abilities to regulate translation rate of ATF4-RLuc in Human embryonic kidney 293T (HEK293T) cells, which were widely used for preliminary screening of ASFV genes based on the genetic reporter systems (13, 25, 32).”; **and please see page 8 lines 170-175:** “Since the natural target cells of ASFV are macrophages, we further investigated the involvement of ASFV MGF110-7L in eIF2 α signaling and ISR in PK-15 cells and porcine alveolar macrophages-derived 3D4/21 cells (36). Both cells were transfected with MGF110-7L expressing plasmids with an increasing dose or empty vector, and treated with TG as positive controls.”

3. In fig 1. please identify the different graphs, some are missing.

Response: We greatly appreciate your valuable comment. Based on your suggestion, we carefully checked the previous Fig. 1 and modified it to include all the essential information. Please see the revised Fig. 1 below.

We sincerely thank the reviewers for their careful and very valuable evaluation of our manuscript. All of the reviewers' suggestions and questions have been addressed.

October 30, 2022

Dr. Shichong Han
College of Veterinary Medicine, Henan Agricultural University
Zhengzhou, Henan 450046
China

Re: Spectrum03282-22R1 (African swine fever virus MGF110-7L induces host cell translation suppression and stress granule formation by activating the PERK/PKR-eIF2 α pathway)

Dear Dr. Shichong Han:

Your manuscript has been accepted, and I am forwarding it to the ASM Journals Department for publication. You will be notified when your proofs are ready to be viewed.

Sincerely,

Manjula Kalia
Editor, Microbiology Spectrum
